# Posterior Mismatch Matters: Adversarial Training for Long-Tailed Robustness

Lilin Zhang [1]   Yue Li [2]   Jiancheng Shi [3]   Jiancheng Lv [1]   Xianggen Liu [1]

## Abstract

Adversarial training breaks down in long-tailed settings, exhibiting severe robustness degradation on worst-performing (often tail) classes. We identify a key cause of this failure as a posterior mismatch: coarse-grained absolute labels collapse class posteriors into point estimates, leading to biased class-frequency estimation and an enlarged robust generalization gap, which ultimately amplifies worst-class vulnerability. To address this issue, we propose Posterior-driven Adversarial Training (PAT), which learns a posterior surrogate to provide fine-grained probabilistic supervision for adversarial training, and integrates weight perturbations to encourage a flatter loss landscape. Our theory shows that accurate posterior approximation simultaneously tightens class-frequency estimation error and robust generalization bounds, while a flat weight loss landscape stabilizes sensitivity to posterior approximation errors. Extensive experiments on long-tailed benchmarks confirm that PAT consistently improves robustness, with especially large gains on worst-class.

## 1. Introduction

Deep neural networks have achieved remarkable success across many domains (Zhang et al., 2026b; Bai et al., 2021). But they remain highly vulnerable to adversarial examples, which are inputs perturbed by imperceptible noise that induce incorrect predictions (Goodfellow et al., 2015). Among existing defenses, adversarial training is widely regarded as the most effective approaches. It formulates learning as a min-max problem, in which adversarial examples are generated to maximize the classification loss while the model is optimized to minimize it, thereby improving robustness

[1]The Collage of Computer Science, Sichuan University, Chengdu, China [2]Dongfang Electric (Chengdu) Innovation Research Co., Ltd., Chengdu, China [3]Southwest China Research Institute of Electronic Equipment, Chengdu, China. Correspondence to: Xianggen Liu <liuxianggen@scu.edu.cn>.

*Proceedings of the 43rd International Conference on Machine Learning*, Seoul, South Korea. PMLR 306, 2026. Copyright 2026 by the author(s).

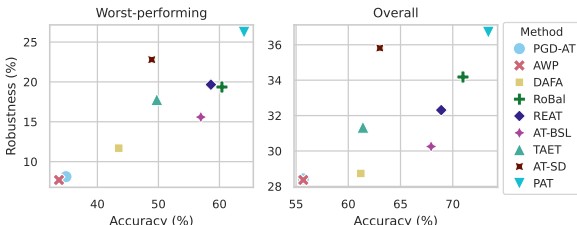

*Figure 1.* Comparison of accuracy and robustness against PGD attack on CIFAR10-LT with imbalance ratio 50 using ResNet-18. Results are averaged over worst-class (left) and all classes (right). PAT consistently outperforms competing methods.

against worst-case perturbations (Bai et al., 2021; Zhao et al., 2022).

Despite its success, adversarial training has predominantly been evaluated on class-balanced benchmarks such as CIFAR-10 and CIFAR-100 (Krizhevsky et al., 2009), which fail to reflect the long-tailed distributions common in real-world applications (Lin et al., 2017; Cao et al., 2019). In long-tailed settings, a small number of head classes dominate the data, while tail classes are severely underrepresented. As a result, models tend to be biased toward head classes and exhibit substantially degraded generalization on tail classes (Wang et al., 2017; Buda et al., 2018). This issue is particularly critical in adversarial scenarios, where attackers are unconstrained by class frequencies and can deliberately target vulnerable tail classes. Consequently, robustness on worst-performing (often tail) classes becomes a more practical and security-relevant evaluation criterion.

However, long-tailed adversarial training remains insufficiently understood. Existing approaches (Wu et al., 2021; Yue et al., 2024; Cho et al., 2025; Yu-Hang et al., 2025; Zhang et al., 2026a) typically combine adversarial training with imbalance-aware objectives such as Balanced Softmax Loss (BSL) (Ren et al., 2020). While partially effective, these methods rely on coarse-grained absolute labels that assign each sample to a single class with full confidence. Such labeling induce a posterior mismatch by collapsing the class posterior into a point estimate, thereby discarding informative uncertainty among competing classes. This mismatch is especially problematic in long-tailed and adversarial settings. On the one hand, in long-tailed scenarios, class-frequency priors are central to rebalancing techniques but highly sensitive to label representations. On the other

hand, prior studies show that improper label handling in adversarial training can exacerbate robust overfitting (Dong et al., 2022; Li et al., 2024b). Taken together, these observations suggest that absolute labels are ill-suited when long-tail imbalance and adversaries coexist.

To gain deeper insight, we theoretically analyze the adverse effects induced by this mismatch and derive principled guidelines to address them. We introduce Bayesian labels, defined as class posterior probabilities, to formalize the discrepancy induced by absolute labeling. Based on this formulation, our analysis establishes that: (i) Compared to Bayesian labels, absolute labels induce more biased class-frequency estimation in long-tailed settings and enlarge the robust generalization gap in adversarial settings, leading to a disproportionate degradation of robustness on the worst-performing class (hereafter worst-class). (ii) Crucially, accurately approximating class posteriors—especially for the worst-class—can simultaneously reduce class-frequency estimation error and the robust generalization gap. Moreover, we show that the flatness of weight loss landscape plays a complementary role: a flatter weight loss landscape not only improves robust generalization (Wu et al., 2020), but also stabilizes the sensitivity of robust generalization gap to posterior approximation errors.

Guided by these insights, we propose Posterior-driven Adversarial Training (PAT). PAT introduces a learnable posterior surrogate to approximate the class posteriors, and uses it to generate fine-grained probabilistic supervision for adversarial training. This posterior surrogate is optimized using a LogSumExp-based objective over class-specific losses on clean data, with emphasis on the worst-class. During adversarial training, PAT leverages these fine-grained labels and incorporates weight perturbations to encourage a flatter weight loss landscape, thereby improving long-tailed robustness. Extensive experiments show that PAT consistently improves robustness, especially on the worst-class, substantially outperforming existing methods as shown in Fig. 1. Our contributions are summarized as follows:

- We identify a posterior mismatch in existing long-tailed adversarial training: coarse-grained absolute labels collapse class posteriors, resulting in biased class-frequency estimation and degraded robust generalization.
- We theoretically show that accurate posterior approximation reduces both class-frequency estimation error and the robust generalization gap, while flatter weight loss landscapes stabilize the impact of approximation errors.
- We propose Posterior-driven Adversarial Training (PAT), which learns a posterior surrogate for fine-grained supervision and incorporates weight perturbations to promote a flatter loss landscape. Extensive experiments demonstrate consistent robustness gains, with particularly strong improvements on tail and worst-performing classes.

## 2. Related works

The problem of adversarial robustness under long-tailed distributions has received limited attention. A few works (Cho et al., 2025; Wu et al., 2021; Yu-Hang et al., 2025; Yue et al., 2024) explore this issue, with most approaches building upon Balanced Softmax Loss (BSL) (Ren et al., 2020). For example, RoBal (Wu et al., 2021) extends BSL with a cosine classifier, class-aware margins, and KL-divergence regularization. However, ablation studies indicate that BSL is the primary contributor to performance gains. AT-BSL (Yue et al., 2024) simplifies this by combining adversarial training directly with BSL, further enhancing robustness through data augmentation. More recently, (Cho et al., 2025) introduces a self-distillation framework to AT-BSL, where a robust and balanced teacher is used to guide the target model by adversarial robust distillation (Goldblum et al., 2020). Additionally, identifying that BSL-based methods tend to struggle with underrepresented classes and are prone to robust overfitting, (Yu-Hang et al., 2025) proposes TAET, a two-stage training strategy consisting of an initial stabilization phase followed by stratified equalization adversarial training. Recently, (Zhang et al., 2026a) proposes a plug-and-play framework RobustLT adjusting the adversarial perturbations by class-wise and iteration-wise control. Further discussion of related topics is provided in Appendix A.

## 3. Preliminaries and problem analysis

### 3.1. Absolute, Bayesian, and predicted labels

Let $p$ be a distribution over $\mathcal{X} \times \mathcal{Y}$, where $\mathcal{X} \subset \mathbb{R}^d$ and $\mathcal{Y} = \{y_i\}_{i \in [C]}$ are the data space and class space, respectively. $S = \{(x^{(k)}, y^{(k)})\}_{k \in [N]}$ is a dataset of $N$ samples drawn i.i.d. from $p$. In standard supervised learning, each sample $x$ is assigned an absolute label $y = \arg\max_{y_i \in \mathcal{Y}} p(y_i|x)$, which induces a one-hot class-probability vector:

$$\mathbf{e}_y = [\mathbb{1}\{y_i = y\}]_{i \in [C]} \in \{0, 1\}^C. \qquad (1)$$

This labeling scheme assumes that each sample belongs exclusively to a single class. Absolute labels answer 'which class', but Bayesian labels answer 'how confident'. The Bayesian label of $x$ is defined as the class posterior vector:

$$\mathbf{y} = [p(y_i|x)]_{i \in [C]} \in [0, 1]^C, \qquad (2)$$

which captures inter-class uncertainty. While $\mathbf{e}_y$ is an unbiased estimator of $\mathbf{y}$ in expectation (Menon et al., 2021), it discards all probabilistic information beyond the most likely class, resulting in a substantial loss of granularity. Given a classifier $h : \mathcal{X} \to \mathbb{R}^C$, applying the softmax function $\sigma[\cdot]$ to the output logits yields the predicted label:

$$\mathbf{y}_h = \sigma[h(x)] = \left[ \frac{\exp(h(x)_i)}{\sum_{y_j \in \mathcal{Y}} \exp(h(x)_j)} \right]_{i \in [C]}, \qquad (3)$$

with predicted class $y_h = \arg\max_{y_i \in \mathcal{Y}} p_h(y_i|x)$.

## 3.2. Adversarial robustness

Let $\mathcal{H}$ denote a hypothesis class of classifiers and $\ell : \mathbb{R}^C \times \mathbb{R}^C \to \mathbb{R}$ be a non-negative loss function to measure the classification error. We focus on the standard cross-entropy loss $\ell(\mathbf{y}_1, \mathbf{y}_2) = -\mathbf{y}_1^\top \log \mathbf{y}_2 = \sum_{i \in [C]} -\mathbf{y}_{1,i} \log \mathbf{y}_{2,i}$, where $\mathbf{y}_{1,i}$ and $\mathbf{y}_{2,i}$ are the $i$-th components of $\mathbf{y}_1$ and $\mathbf{y}_2$.

Adversarial robustness measures the ability of a classifier $h \in \mathcal{H}$ to correctly predict the label of an adversarial example $\tilde{x}$, which is generated by perturbing a clean sample $x$ within a small neighborhood to cause misclassification. Under the standard assumption that adversarial perturbations preserve semantic labels, i.e., $\mathbf{e}_{\tilde{y}} = \mathbf{e}_y$. We consider the widely adopted $l_\infty$-attack with perturbation budget $\epsilon$, under which an adversarial example is defined as:

$$\tilde{x} = \underset{\tilde{x} : \|\tilde{x} - x\|_\infty \leq \epsilon}{\arg\max} \ \ell(\mathbf{e}_y, \tilde{\mathbf{y}}_h), \tag{4}$$

where $\tilde{\mathbf{y}}_h$ denotes the predicted label of $\tilde{x}$. The robustness of $h$ is traditionally reflected by robust generalization error under distribution $p$ defined as $\mathcal{R}(h, p) = \mathbb{E}_{(x,y) \sim p}[\ell(\mathbf{e}_y, \tilde{\mathbf{y}}_h)]$. Since absolute label is an unbiased estimator of the Bayesian label, which gives $\mathbb{E}_{y|x \sim p}[\mathbf{e}_y] = \mathbf{y}$, this quantity can equivalently be written as:

$$\mathcal{R}(h, p) = \mathbb{E}_{(x, \cdot) \sim p}[\ell(\mathbf{y}, \tilde{\mathbf{y}}_h)]. \tag{5}$$

### 3.3. Posterior mismatch affects long-tailed robustness

In long-tailed settings, severe class imbalance $p(y_1) \geq p(y_2) \geq \cdots \geq p(y_C)$ results in a class-imbalanced training set $S \sim p^N$. As a consequence, direct adversarial training objectives over $S$ tend to be dominated by head classes, thereby underestimating the vulnerability of tail classes.

Most existing long-tailed adversarial training approaches are based on BSL. Given an input $x$, BSL modifies the predict label for $x$ with class-frequency-dependent margins as:

$$\mathbf{y}_h^{\mathrm{BSL}} = \sigma\big[h(x) + \tau_b \log \mathbf{f}\big], \tag{6}$$

where $\mathbf{f} = \mathbb{E}_{(x,y) \in S}[\mathbf{e}_y]$ denotes the class-frequency prior[1], and $\tau_b$ controls the margin strength for each class. Traditionally, long-tailed adversarial training first obtains adversarial data $\tilde{x}$ by maximizing standard cross-entropy loss (Eq. (4)), then optimizes the model by minimizing:

$$\mathcal{L}_{\mathrm{BSL}}(h, S) = \mathbb{E}_{(x,y) \in S}[\ell(\mathbf{e}_y, \tilde{\mathbf{y}}_h^{\mathrm{BSL}})], \tag{7}$$

where $\tilde{\mathbf{y}}_h^{\mathrm{BSL}}$ is the BSL modified predict label for $\tilde{x}$. Together, Eq. (4) and Eq. (7) form the standard min-max optimization framework for adversarial training.

Although effective to some extent, their reliance on coarse-grained absolute labels is problematic when long-tailed imbalance and adversarial perturbations coexist. To gain a

---

[1] $\mathbf{f} = [p(y_i)]_{i \in [C]}$ in population case.

deeper insight, we examine the intrinsic role of the population BSL objective $\mathcal{L}_{\mathrm{BSL}}(h, p) = \mathbb{E}_{(x,y) \sim p}[\ell(\mathbf{e}_y, \tilde{\mathbf{y}}_h^{\mathrm{BSL}})] = \mathbb{E}_{(x, \cdot) \sim p}[\ell(\mathbf{y}, \tilde{\mathbf{y}}_h^{\mathrm{BSL}})]$. Let $\mathrm{H}(\cdot)$ and $\mathrm{MI}(\cdot, \cdot)$ denote entropy and mutual information, respectively (Alajaji et al., 2018).

**Lemma 3.1** (Nature of BSL). *When $\tau_b = 1$, $\mathcal{L}_{BSL}(h, p) \geq H(y) - MI(\tilde{x}, y) = H(y|\tilde{x})$.*

*Remark* 3.2. Lemma 3.1 shows that population BSL objective encourages the classifier to capture mutual information between adversarial input $\tilde{x}$ and class $y$. By incorporating class-frequency information, BSL rebalances the training process, facilitating more class-balanced robust generalization on distribution $p$, which is critical for obtaining a lower $\mathcal{R}(h, p)$. It also suggests $\tau_b = 1$ as an optimal choice, consistent with empirical findings in (Yue et al., 2024).

However, the desirable property in Lemma 3.1 implicitly relies on access to the Bayesian labels as $\mathbb{E}_{y|x \sim p}[\mathbf{e}_y] = \mathbf{y}$. We will show that using absolute labels in the empirical objective leads to non-negligible negative effects when long-tailed data imbalance and adversarial perturbations coexist.

# 4. Theoretical insights

Here, we analyze the consequences of the posterior mismatch identified in Section 3.3 in empirical long-tailed adversarial settings, and derive insights that motivate the design of our method. All proofs are deferred to Appendix B.

## 4.1. Negative effects of posterior mismatch

**Effect 1: Biased class-frequency estimation.** Let $\mathrm{Var}[\cdot]$ denote the variance[2]. We first show that Bayesian labels exhibit lower class-probability variance than absolute labels.

**Lemma 4.1** (Class-probability variance). $\mathrm{Var}[\mathbf{y}_i] \leq \mathrm{Var}[\mathbf{e}_{y,i}]$. *Equality holds only when $\mathbf{y}_i = \mathbf{e}_{y,i}$ over $p$.*

We next characterize how the posterior mismatch affects class-frequency estimation in long-tail scenarios.

**Theorem 4.2** (Class-frequency bias). *For $\forall y_i \in \mathcal{Y}$ and $\delta \in (0, 1)$, each of the following inequalities holds with probability at least $1 - \delta$:*
*(i) $|\mathbb{E}_{(x, \cdot) \in S}[\mathbf{y}_i] - p(y_i)| \leq \sqrt{\mathrm{Var}[\mathbf{y}_i]} A_1 + A_2$;*
*(ii) $|\mathbb{E}_{(x,y) \in S}[\mathbf{e}_{y,i}] - p(y_i)| \leq \sqrt{\mathrm{Var}[\mathbf{e}_{y,i}]} A_1 + A_2$,*
*where $A_1 = \sqrt{\frac{2 \log(2C/\delta)}{N}}$ and $A_2 = \frac{\log(2C/\delta)}{3N}$.*

*Remark* 4.3. By Lemma 4.1, absolute labels induce strictly larger variance than Bayesian labels. Consequently, class-frequency estimates obtained from absolute labels deviate more from the true class-frequency according to Theorem 4.2. When such biased estimates are used in rebalance techniques like BSL, they can lead to either under- or over-compensating for class imbalance, ultimately harming the rebalancing effect and degrading worst-class generalization.

---

[2] $\mathrm{Var}[z] = \mathbb{E}[z^2] - (\mathbb{E}[z])^2$

**Effect 2: Enlarged robust generalization gap.** Beyond frequency bias, we analyze how the mismatch affects robust generalization in adversarial settings. Since prior work observes a strong connection between the weight loss landscape and the robust generalization gap (Wu et al., 2020), we further consider the landscape of robust error $\mathcal{R}(h, p)$ (Eq. (5)) defined as:

$$\mathbb{E}_{u \in \mathcal{U}}\big[\,\mathcal{R}(h + u, p)\big], \qquad (8)$$

where $u$ is a perturbation from a predefined weight neighborhood $\mathcal{U} = \{u : \|u\|_2 \leq \gamma\|h\|_2\}$ with intensity $\gamma$. We focus on the gap between this landscape and the empirical objective $\mathcal{L}_{\text{BSL}}(h, S)$ (Eq. (7)). First, we show that Bayesian labels induce smaller loss variance than absolute labels.

**Lemma 4.4** (Loss variance). *For any $h \in \mathcal{H}$, $\text{Var}[\ell(\mathbf{y}, \tilde{\mathbf{y}}_h^{BSL})] \leq \text{Var}[\ell(\mathbf{e}_y, \tilde{\mathbf{y}}_h^{BSL})]$ and the equality holds only if $\ell(\mathbf{e}_y, \tilde{\mathbf{y}}_h^{BSL})$ is a constant over $p$.*

Let $\Delta_{h+u} = \max_{x \in \mathcal{X}, i \in [C]}[-\log \tilde{\mathbf{y}}_{h+u,i}^{\text{BSL}})]$ denote a uniform upper bound on the loss under weight perturbation $u$. We then obtain the following generalization bound.

**Theorem 4.5** (Robust generalization gap). *For $\delta \in (0, 1)$, provided that $\mathcal{L}_{BSL}(h, p) \geq \mathcal{R}(h, p)$, the inequality holds with a probability at least $1 - \delta$:*

$$\mathbb{E}_{u \in \mathcal{U}}[\mathcal{R}(h + u, p)] - \mathcal{L}_{BSL}(h, S)$$
$$\leq \mathbb{E}_{u \in \mathcal{U}}\left[\sqrt{\text{Var}[\ell(\mathbf{e}_y, \tilde{\mathbf{y}}_{h+u}^{BSL})]}\, C_1 + C_2\right] \text{ (deviation)}$$
$$+ \mathbb{E}_{u \in \mathcal{U}}[\mathcal{L}_{BSL}(h + u, S) - \mathcal{L}_{BSL}(h, S)] \text{ (sharpness)},$$

*where $C_1 = \sqrt{\frac{2\log(|\mathcal{U}|/\delta)}{N}}$, $C_2 = \Delta_{h+u}\frac{\log(|\mathcal{U}|/\delta)}{3N}$.*

*Remark* 4.6. Theorem 4.5 reveals two dominant contributors to the robust generalization gap: (i) a deviation error proportional to the loss variance, and (ii) a sharpness penalty that captures sensitivity to weight perturbations. Since absolute labels incur larger loss variance (cf. Lemma 4.4), they increase the deviation error and are also associated with sharper loss landscapes. As existing long-tailed adversarial training methods using absolute labels do not explicitly control sharpness, they tend to suffer from larger gaps.

In summary, the posterior mismatch causes: (i) biased class-frequency estimation in long-tailed settings and (ii) enlarged robust generalization gap in adversarial settings. Together, these two negative effects harm long-tailed robustness.

### 4.2. Principled guidelines for solutions

To mitigate the effects, we introduce a posterior surrogate $q$, which models a distribution $q(y|x)$ to approximate the class posterior $p(y|x)$. Using $q$ to generate probabilistic supervision yields a soft-labeled dataset $S_q = \{(x, \mathbf{y}_q) \text{ s.t. } x \in S, \mathbf{y}_q = [q(y_i|x)]_{i \in [C]}\}$. We then analyze the properties that $q$ should satisfy for effective long-tailed adversarial training.

**Guideline 1: A surrogate accurate even on worst-class.**

**Theorem 4.7** (Class-frequency bias with $q$). *For $\forall y_i \in \mathcal{Y}$ and $\delta \in (0, 1)$, it holds with probability at least $1 - \delta$:*

$$|\mathbb{E}_{(x, \cdot) \in S}[\mathbf{y}_{q,i}] - p(y_i)| \leq \sqrt{\text{Var}[\mathbf{y}_{q,i}]}A_1 + A_2 \text{ (deviation)}$$
$$+ \mathbb{E}_{x \sim p}[|\mathbf{y}_{q,i} - \mathbf{y}_i|] \text{ (approximation)},$$

*where $A_1 = \sqrt{\frac{2\log(2C/\delta)}{N}}$ and $A_2 = \frac{\log(2C/\delta)}{3N}$.*

*Remark* 4.8. Compared to Theorem 4.2, the bound with $q(y|x)$ introduces an explicit approximation error while retaining the deviation term of the same structure. As $q(y|x)$ becomes more accurate on class $y_i$, the approximation term vanishes and $\text{Var}[\mathbf{y}_{q,i}]$ approaches $\text{Var}[\mathbf{y}_i]$, yielding a strictly tighter frequency bound than that obtained with absolute labels (cf. Theorem 4.2).

Theorem 4.7 naturally motivates a class-wise perspective: only when the posterior surrogate is accurate across all classes can class-frequency estimates remain unbiased. In long-tailed settings, this requires the surrogate to be accurate even on the worst-class, which constitutes Guideline 1.

**Guideline 2: A flat weight loss landscape.**

**Theorem 4.9** (Robust generalization gap with $q$). *For $\delta \in (0, 1)$, provided that $\mathcal{L}_{BSL}(h, q) \geq \mathcal{R}(h, q)$, the following inequality holds with probability at least $1 - \delta$:*

$$\mathbb{E}_{u \in \mathcal{U}}[\mathcal{R}(h + u, p)] - \mathcal{L}_{BSL}(h, S_q)$$
$$\leq \mathbb{E}_{u \in \mathcal{U}}\left[\sqrt{\text{Var}[\ell(\mathbf{y}_q, \tilde{\mathbf{y}}_{h+u}^{BSL})]}\, C_1 + C_2\right] \text{ (deviation)}$$
$$+ \mathbb{E}_{u \in \mathcal{U}}[\mathcal{L}_{BSL}(h + u, S_q) - \mathcal{L}_{BSL}(h, S_q)] \text{ (sharpness)}$$
$$+ \mathbb{E}_{u \in \mathcal{U}}[\Delta_{h+u}] \sum_{i \in [C]} \mathbb{E}_{(x, \cdot) \sim p}[|\mathbf{y}_{q,i} - \mathbf{y}_i|] \text{ (approximation)},$$

*where $C_1 = \sqrt{\frac{2\log(|\mathcal{U}|/\delta)}{N}}$, $C_2 = \Delta_{h+u}\frac{\log(|\mathcal{U}|/\delta)}{3N}$.*

*Remark* 4.10. Compared to Theorem 4.5, the bound with $q$ introduces an explicit approximation term while preserving the same deviation and sharpness structure. When $q(y|x)$ accurately approximates $p(y|x)$ across all classes (as required by Guideline 1) the approximation error becomes negligible and the sample variance is reduced (cf. Lemma 4.4), yielding a tighter bound. Crucially, a flatter weight loss landscape not only reduces the sharpness term, but also stabilizes the impact of approximation error by keeping its coefficient $\mathbb{E}_{u \in \mathcal{U}}[\Delta_{h+u}]$ close to $\Delta_h$. This interaction further tightens the robust generalization bound and motivates Guideline 2. Moreover, the dependence of $C_1$ and $C_2$ on the sample size $N$ offers a theoretical rationale for why data augmentation often improves long-tailed robustness (Yue et al., 2024). This further suggests that once posterior mismatch is reduced, the gap becomes less sensitive to $N$, diminishing the marginal benefit of data augmentation.

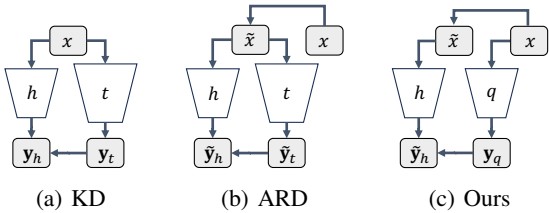

(a) KD          (b) ARD          (c) Ours

*Figure 2.* Concept diagrams of Knowledge Distillation (KD) (Hinton et al., 2014), Adversarial Robustness Distillation (ARD) (Goldblum et al., 2020), and our formulation. Here, $x$ and $\tilde{x}$ are clean and adversarial data, respectively. The labels $\mathbf{y}_h, \mathbf{y}_t$, and $\mathbf{y}_q$ are produced by target model $h$, teacher model $t$, and posterior surrogate $q$ for input $x$, respectively. The labels $\tilde{\mathbf{y}}_h$ and $\tilde{\mathbf{y}}_t$ are for $\tilde{x}$.

Generally, Theorems 4.7 and 4.9 formalize how mitigating the posterior mismatch (especially on worst-class) combined with a flat weight loss landscape yields tighter guarantees for long-tailed adversarial robustness. Although our formulation bears superficial resemblance to distillation techniques, the objectives are fundamentally different, as illustrated in Fig. 2. Further discussion is provided in Appendix A.2.

## 5. Methodology

In this section, we present Posterior-driven Adversarial Training (PAT), a principled framework directly motivated by the theoretical guidelines in Section 4.2.

### 5.1. Learnable approximation of class posterior

Motivated by Guideline 1, we introduce a posterior surrogate $q$ that accurately approximates the class posterior including the worst-class. Here, $q$ is implemented as a neural network followed by a softmax function $\sigma[\cdot]$ (Eq. (3)), and is trained to approximate $p(y|x)$ with explicit emphasis on worst-class performance. Importantly, the posterior mismatch we identify arises when absolute labels are directly used to adversarially train the target model $h$. Here, where only absolute labels are available, they will be used solely as empirical supervision to learn a posterior surrogate.

According to Lemma 3.1, optimizing the BSL objective ($\tau_b = 1$) over clean data can encourage surrogate $q$ to capture the mutual information between input $x$ and class $y$, leading to a better approximation of $p(y|x)$. This objective is defined as:

$$\mathcal{L}(q, S) = \mathbb{E}_{(x,y) \in S}[-\mathbf{e}_y^\top \log \sigma[q(x) + \log \mathbf{f}]], \quad (9)$$

where $\mathbf{f} = \mathbb{E}_{(x,y) \in S}[\mathbf{e}_y]$ is the prior class-frequency in the dataset. To explicitly focus on worst-class behavior, we define the class-specific loss for each $y_i \in \mathcal{Y}$ as:

$$\mathcal{L}_{y_i}(q, S) = \mathbb{E}_{(x,y) \in S}[-\mathbf{e}_{y,i} \log \sigma[q(x) + \log \mathbf{f}]_i], \quad (10)$$

where $\sigma[\cdot]_i$ denote the i-th softmax component. We then aggregate these class-wise losses using a LogSumExp (LSE)

operator, a smooth approximation to the max:

$$\mathcal{L}_{\text{LSE}}(q, S) = \tau \log \sum_{y_i \in \mathcal{Y}} \exp(\mathcal{L}_{y_i}(q, S)), \quad (11)$$

where $\tau$ controls the tightness of the approximation[3]. The posterior surrogate is obtained by $q = \arg\min_q \mathcal{L}_{\text{LSE}}(q, S)$, and then kept fixed during adversarial training phase, without further updates or refreshing.

### 5.2. Fine-grained labels and class-frequencies

After learning the posterior surrogate $q$, we use it to generate fine-grained labels for adversarially training the target model $h$. For each sample $(x, y) \in S$, the adversarial example $\tilde{x}$ is generated by Eq. (4) using PGD (Madry et al., 2018):

$$\tilde{x} \leftarrow \text{project}_\epsilon\{\tilde{x} + \alpha \, \text{sign}(\nabla_{\tilde{x}} \ell(\mathbf{e}_y, \tilde{\mathbf{y}}_h))\}, \quad (12)$$

where $\alpha$ is the step size, $\text{project}_\epsilon$ ensures that the perturbation remains within the $l_\infty$-norm constraint $\|\tilde{x} - x\|_\infty \leq \epsilon$, and $\tilde{x}$ is initialized as $x$ with a small random perturbation.

Intuitively, while the absolute label remains unchanged under perturbations, the underlying class-probability may shift leading to a different Bayesian label $\tilde{\mathbf{y}} \neq \mathbf{y}$. To capture this fine-grained shift, we leverage $q$ to predict soft labels (by softmax $\sigma$) for both clean and adversarial samples and approximate $\tilde{\mathbf{y}}$ by constructing a mixed label as:

$$\hat{\mathbf{y}}_q = (1 - r) \cdot \sigma[q(x)] + r \cdot \sigma[q(\tilde{x})], \quad (13)$$

where $r \in [0, 1]$ controls the mixing ratio. $\sigma[q(x)]$ serves as a stable reference that enforces prediction consistency between clean and adversarial data, while $\sigma[q(\tilde{x})]$ captures local posterior shift caused by adversarial perturbation. This combination enables the model to better account for robustness in long-tailed scenarios under posterior mismatch.

To further mitigate the bias in class-frequency estimation, class-frequency will be updated by:

$$\hat{\mathbf{f}}_q \leftarrow \mathbb{E}_{(x,\cdot) \in S}[\hat{\mathbf{y}}_q]. \quad (14)$$

This class-frequency update also avoids a distributional inconsistency in training objective, which could otherwise lead to unstable training. Optimizing the BSL objective with respect to a data-label distribution encourages the model to capture the mutual information between data and label as shown in Lemma 3.1. Therefore, training with the fine-grained class-frequency distributionally consistent with the fine-grained labels ensures a stable optimization objective.

The resulting adversarial training objective is defined as:

$$\mathcal{L}_{\text{PAT}}(h, S_q) = \mathbb{E}_{(x,\cdot) \in S}[-\hat{\mathbf{y}}_q^\top \log(\sigma[h(\tilde{x}) + \log \hat{\mathbf{f}}_q])]. \quad (15)$$

---

[3]For $\{z_i\}_{i \in [C]}$, $\max_{i \in [C]} z_i \leq \tau \log \sum_{i=1}^{C} \exp(z_i/\tau) \leq \max_{i \in [C]} z_i + \tau \log C$ (Samakhoana & Grimmer, 2025). Since $\mathcal{L}(q, S) = \sum_{i \in [C]} \mathcal{L}_{y_i}(q, S)$ and each class-specific loss scales inversely with $C$, only a coefficient $\tau$ proportional to $C$ is needed.

*Table 1.* Accuracy and robustness of various algorithms on CIFAR10-LT. The $1^{st}$ and $2^{nd}$ results are highlighted.

| Method | Acc. (ResNet-18) | | | Rob. (ResNet-18) | | | Acc. (WRN-28-10) | | | Rob. (WRN-28-10) | | |
|---|---|---|---|---|---|---|---|---|---|---|---|---|
| | all | tail | worst | all | tail | worst | all | tail | worst | all | tail | worst |
| AT | $55.70_{\pm0.85}$ | $45.40_{\pm1.05}$ | $34.85_{\pm1.21}$ | $28.42_{\pm0.24}$ | $15.21_{\pm0.39}$ | $8.11_{\pm0.56}$ | $57.91_{\pm0.28}$ | $48.18_{\pm0.34}$ | $36.18_{\pm0.24}$ | $27.26_{\pm0.14}$ | $13.75_{\pm0.20}$ | $7.45_{\pm0.13}$ |
| AWP | $55.69_{\pm0.37}$ | $45.26_{\pm0.46}$ | $33.71_{\pm0.85}$ | $28.36_{\pm0.28}$ | $14.83_{\pm0.50}$ | $7.70_{\pm0.49}$ | $58.75_{\pm0.04}$ | $49.00_{\pm0.13}$ | $37.37_{\pm0.39}$ | $28.47_{\pm0.40}$ | $14.65_{\pm0.52}$ | $8.60_{\pm0.40}$ |
| DAFA | $61.19_{\pm0.90}$ | $52.73_{\pm1.17}$ | $43.49_{\pm1.79}$ | $28.73_{\pm0.37}$ | $17.74_{\pm0.73}$ | $11.69_{\pm1.00}$ | $61.51_{\pm0.16}$ | $52.99_{\pm0.14}$ | $41.70_{\pm0.07}$ | $27.44_{\pm0.12}$ | $15.22_{\pm0.17}$ | $9.84_{\pm0.54}$ |
| RoBal | $\underline{70.98}_{\pm0.21}$ | $\underline{65.29}_{\pm0.38}$ | $\underline{60.41}_{\pm0.19}$ | $34.18_{\pm0.54}$ | $25.10_{\pm0.82}$ | $19.36_{\pm1.48}$ | $\underline{71.62}_{\pm1.22}$ | $\underline{65.62}_{\pm1.61}$ | $\underline{61.11}_{\pm2.36}$ | $32.02_{\pm0.14}$ | $21.05_{\pm0.62}$ | $16.19_{\pm1.36}$ |
| REAT | $68.90_{\pm0.32}$ | $63.07_{\pm0.41}$ | $58.60_{\pm0.88}$ | $32.31_{\pm0.61}$ | $23.72_{\pm1.08}$ | $19.66_{\pm0.97}$ | $67.69_{\pm0.57}$ | $60.68_{\pm0.72}$ | $54.96_{\pm1.61}$ | $28.97_{\pm0.32}$ | $17.18_{\pm0.50}$ | $12.69_{\pm0.31}$ |
| AT-BSL | $67.93_{\pm0.73}$ | $61.36_{\pm1.03}$ | $56.95_{\pm1.80}$ | $30.25_{\pm0.05}$ | $19.98_{\pm0.44}$ | $15.59_{\pm1.19}$ | $64.11_{\pm0.37}$ | $56.16_{\pm0.53}$ | $47.70_{\pm1.01}$ | $27.77_{\pm0.18}$ | $15.10_{\pm0.15}$ | $10.02_{\pm0.48}$ |
| TAET | $61.40_{\pm3.68}$ | $56.24_{\pm3.83}$ | $49.73_{\pm5.45}$ | $31.34_{\pm0.25}$ | $24.40_{\pm0.44}$ | $17.79_{\pm0.94}$ | $65.70_{\pm2.12}$ | $60.98_{\pm2.24}$ | $55.25_{\pm2.89}$ | $31.84_{\pm0.28}$ | $24.96_{\pm0.82}$ | $\underline{20.26}_{\pm1.07}$ |
| AT-SD | $63.02_{\pm0.35}$ | $58.48_{\pm0.60}$ | $48.91_{\pm0.25}$ | $\underline{35.82}_{\pm0.26}$ | $\underline{31.03}_{\pm0.71}$ | $\underline{22.79}_{\pm0.29}$ | $65.85_{\pm0.38}$ | $59.40_{\pm0.63}$ | $54.47_{\pm0.77}$ | $\underline{34.96}_{\pm0.38}$ | $\underline{26.03}_{\pm1.20}$ | $19.70_{\pm1.26}$ |
| **PAT** | $\mathbf{73.41}_{\pm0.55}$ | $\mathbf{69.31}_{\pm0.58}$ | $\mathbf{64.03}_{\pm0.17}$ | $\mathbf{36.71}_{\pm0.18}$ | $\underline{30.31}_{\pm0.11}$ | $\mathbf{26.23}_{\pm0.42}$ | $\mathbf{76.59}_{\pm0.34}$ | $\mathbf{72.64}_{\pm0.44}$ | $\mathbf{69.17}_{\pm0.54}$ | $\mathbf{36.70}_{\pm0.34}$ | $\mathbf{28.93}_{\pm0.68}$ | $\mathbf{25.93}_{\pm0.83}$ |

*Table 2.* Accuracy and robustness of various algorithms on CIFAR100-LT. The $1^{st}$ and $2^{nd}$ results are highlighted.

| Method | Acc. (ResNet-18) | | | Rob. (ResNet-18) | | | Acc. (WRN-28-10) | | | Rob. (WRN-28-10) | | |
|---|---|---|---|---|---|---|---|---|---|---|---|---|
| | all | tail | worst | all | tail | worst | all | tail | worst | all | tail | worst |
| AT | $41.73_{\pm0.13}$ | $37.45_{\pm0.16}$ | $23.74_{\pm0.67}$ | $16.59_{\pm0.29}$ | $14.67_{\pm0.41}$ | $5.05_{\pm0.47}$ | $44.25_{\pm0.25}$ | $39.88_{\pm0.33}$ | $26.63_{\pm0.07}$ | $16.88_{\pm0.24}$ | $14.75_{\pm0.29}$ | $4.89_{\pm0.34}$ |
| AWP | $42.28_{\pm0.15}$ | $37.75_{\pm0.20}$ | $24.19_{\pm0.16}$ | $16.38_{\pm0.21}$ | $14.22_{\pm0.26}$ | $4.78_{\pm0.24}$ | $45.66_{\pm0.21}$ | $41.02_{\pm0.18}$ | $27.13_{\pm0.50}$ | $18.24_{\pm0.07}$ | $15.84_{\pm0.24}$ | $5.71_{\pm0.41}$ |
| DAFA | $42.31_{\pm0.12}$ | $38.21_{\pm0.20}$ | $24.99_{\pm0.56}$ | $16.33_{\pm0.25}$ | $14.68_{\pm0.16}$ | $5.29_{\pm0.41}$ | $44.59_{\pm0.31}$ | $40.72_{\pm0.30}$ | $27.80_{\pm0.85}$ | $16.80_{\pm0.19}$ | $15.08_{\pm0.25}$ | $5.60_{\pm0.30}$ |
| RoBal | $\underline{46.54}_{\pm0.46}$ | $\underline{44.55}_{\pm0.45}$ | $\underline{30.49}_{\pm0.76}$ | $19.41_{\pm0.24}$ | $18.53_{\pm0.18}$ | $7.22_{\pm0.07}$ | $\underline{49.22}_{\pm0.38}$ | $\underline{46.80}_{\pm0.27}$ | $\underline{34.46}_{\pm0.33}$ | $18.96_{\pm0.10}$ | $17.89_{\pm0.14}$ | $7.26_{\pm0.15}$ |
| REAT | $44.92_{\pm0.39}$ | $43.00_{\pm0.43}$ | $30.47_{\pm0.46}$ | $17.35_{\pm0.31}$ | $16.56_{\pm0.44}$ | $6.49_{\pm0.37}$ | $47.12_{\pm0.10}$ | $45.29_{\pm0.15}$ | $33.28_{\pm0.45}$ | $17.65_{\pm0.22}$ | $16.90_{\pm0.24}$ | $6.44_{\pm0.10}$ |
| AT-BSL | $44.07_{\pm0.46}$ | $42.01_{\pm0.30}$ | $29.51_{\pm0.50}$ | $16.70_{\pm0.15}$ | $15.75_{\pm0.16}$ | $5.91_{\pm0.10}$ | $46.40_{\pm0.04}$ | $44.00_{\pm0.17}$ | $31.68_{\pm0.25}$ | $17.22_{\pm0.16}$ | $16.00_{\pm0.27}$ | $5.79_{\pm0.18}$ |
| TAET | $44.15_{\pm0.33}$ | $40.20_{\pm0.24}$ | $27.21_{\pm0.48}$ | $18.94_{\pm0.20}$ | $16.74_{\pm0.12}$ | $6.94_{\pm0.12}$ | $45.55_{\pm0.07}$ | $41.15_{\pm0.20}$ | $27.83_{\pm0.45}$ | $18.90_{\pm0.37}$ | $16.59_{\pm0.44}$ | $6.47_{\pm0.30}$ |
| AT-SD | $41.80_{\pm0.45}$ | $40.02_{\pm0.48}$ | $25.12_{\pm0.59}$ | $\mathbf{22.14}_{\pm0.38}$ | $\mathbf{21.64}_{\pm0.39}$ | $\underline{9.15}_{\pm0.22}$ | $44.94_{\pm0.20}$ | $43.57_{\pm0.35}$ | $30.65_{\pm0.98}$ | $\underline{21.46}_{\pm0.22}$ | $\underline{20.90}_{\pm0.36}$ | $\underline{9.37}_{\pm0.18}$ |
| **PAT** | $\mathbf{48.03}_{\pm0.19}$ | $\mathbf{45.97}_{\pm0.12}$ | $\mathbf{31.55}_{\pm0.27}$ | $\underline{22.11}_{\pm0.19}$ | $\underline{21.28}_{\pm0.20}$ | $\mathbf{9.27}_{\pm0.39}$ | $\mathbf{51.25}_{\pm0.33}$ | $\mathbf{49.07}_{\pm0.38}$ | $\mathbf{35.33}_{\pm0.21}$ | $\mathbf{23.39}_{\pm0.33}$ | $\mathbf{22.46}_{\pm0.22}$ | $\mathbf{10.16}_{\pm0.22}$ |

## 5.3. Model update under weight perturbations

Based on Guideline 2, we incorporate adversarial weight perturbations (AWP) (Wu et al., 2020) to enforce local flatness of the weight loss landscape. This modifies the model optimization problem from simply $\min_h \mathcal{L}_{PAT}(h, S_q)$ to $\min_h \max_u \mathcal{L}_{PAT}(h + u, S_q)$. Specifically, the worst-case weight perturbation $\tilde{u}$ is computed as:

$$\tilde{u} \leftarrow \gamma\|h\|_2 \frac{\nabla_h \mathcal{L}_{PAT}(h, S_q)}{\|\nabla_h \mathcal{L}_{PAT}(h, S_q)\|_2}, \quad (16)$$

where $\gamma$ controls the perturbation magnitude. The model weights are then updated under weight perturbation $\tilde{u}$ via:

$$h \leftarrow (h + \tilde{u}) - \eta \nabla_{h+\tilde{u}} \mathcal{L}_{PAT}(h + \tilde{u}, S_q) - \tilde{u}, \quad (17)$$

where $\eta$ is the learning rate.

## 6. Experiments

**Configurations.** We conduct experiments on CIFAR10-LT, CIFAR100-LT, and TinyImageNet-LT, which are long-tailed versions of CIFAR10 (the number of classes $C = 10$), CIFAR100 ($C = 100$) (Krizhevsky et al., 2009), and TinyImageNet ($C = 200$) (Le & Yang, 2015), constructed using the protocol of (Cao et al., 2019). For the main experiments, we set the imbalance ratio to 50 for CIFAR10-LT and 10 for the other datasets. The long-tailed split is generated randomly three times in each setting, and results are reported as averages with standard deviation.s We evaluate the target model across multiple architectures, including ResNet-18 (He et al., 2016a), WideResNet-28-10 (WRN-28-10) (Zagoruyko &

Komodakis, 2016), and PreAct-ResNet-50 (PreAct-RN-50) (He et al., 2016b). The posterior surrogate adopts the same architecture as the model. Hyper-parameter settings are detailed in Section 6.3. Implementation details and additional results are provided in Appendix D, and reproduction information can be found in Appendix C.

**Evaluation protocols.** We report clean accuracy (*Acc.*) to evaluate standard generalization, and robust accuracy (*Rob.*) under $l_\infty$ attacks with perturbation budget $\epsilon = 8/255$ to measure adversarial robustness. Unless otherwise specified, PGD (Madry et al., 2018) with 20 iterations and a step size of $1/255$ is used. To properly reflect performance in long-tailed scenarios, we report clean/robust accuracy averaged over: (i) *all* classes; (ii) *tail* classes, defined as the 80% of classes with the fewest training samples; (iii) *worst* classes, defined as the half of classes with the lowest accuracy. We compare PAT against: (i) standard adversarial training methods, including AT (Madry et al., 2018) and AWP (Wu et al., 2020); (ii) fairness-oriented adversarial training method DAFA (Lee et al., 2024); (iii) long-tail adversarial training methods, including RoBal (Wu et al., 2021), REAT (Li et al., 2024a), AT-BSL (Yue et al., 2024), AT-SD[4] (Cho et al., 2025), and TAET (Yu-Hang et al., 2025).

### 6.1. Main results

The main results are reported in Tables 1 to 3. PAT consistently achieves superior accuracy and robustness across all datasets and architectures. Importantly, the gains are most

---

[4]Results of AT-SD are obtained using our reimplementation.

*Table 3.* Accuracy and robustness of various algorithms on TinyImageNet-LT. The **1st** and 2nd results are highlighted.

| Method | Acc. (ResNet-18) | | | Rob. (ResNet-18) | | | Acc. (PreAct-RN-50) | | | Rob. (PreAct-RN-50) | | |
|---|---|---|---|---|---|---|---|---|---|---|---|---|
| | all | tail | worst | all | tail | worst | all | tail | worst | all | tail | worst |
| AT | $35.78_{\pm0.41}$ | $32.33_{\pm0.41}$ | $20.39_{\pm0.24}$ | $10.51_{\pm0.05}$ | $9.18_{\pm0.06}$ | $2.93_{\pm0.34}$ | $38.20_{\pm0.17}$ | $34.70_{\pm0.15}$ | $22.75_{\pm0.83}$ | $11.77_{\pm0.16}$ | $10.43_{\pm0.21}$ | $3.49_{\pm0.35}$ |
| AWP | $35.95_{\pm0.54}$ | $32.26_{\pm0.53}$ | $20.20_{\pm1.13}$ | $10.36_{\pm0.24}$ | $9.02_{\pm0.24}$ | $2.71_{\pm0.19}$ | $38.75_{\pm0.55}$ | $35.14_{\pm0.58}$ | $22.57_{\pm0.87}$ | $11.76_{\pm0.21}$ | $10.53_{\pm0.20}$ | $3.26_{\pm0.23}$ |
| DAFA | $35.52_{\pm0.44}$ | $32.17_{\pm0.35}$ | $20.32_{\pm0.97}$ | $10.37_{\pm0.09}$ | $9.16_{\pm0.05}$ | $2.91_{\pm0.32}$ | $37.84_{\pm0.23}$ | $34.38_{\pm0.44}$ | $22.55_{\pm0.33}$ | $11.58_{\pm0.08}$ | $10.35_{\pm0.10}$ | $3.63_{\pm0.27}$ |
| RoBal | $\underline{38.92}_{\pm0.47}$ | $\underline{36.32}_{\pm0.47}$ | $23.33_{\pm1.06}$ | $12.73_{\pm0.16}$ | $11.79_{\pm0.28}$ | $4.11_{\pm0.58}$ | $\underline{42.01}_{\pm0.15}$ | $\underline{39.53}_{\pm0.12}$ | $26.45_{\pm0.42}$ | $15.13_{\pm0.22}$ | $14.33_{\pm0.26}$ | $5.56_{\pm0.28}$ |
| REAT | $38.53_{\pm0.22}$ | $36.23_{\pm0.09}$ | $\underline{24.60}_{\pm1.22}$ | $11.66_{\pm0.23}$ | $11.06_{\pm0.22}$ | $3.94_{\pm0.65}$ | $40.52_{\pm0.06}$ | $38.18_{\pm0.10}$ | $\mathbf{26.85}_{\pm0.59}$ | $12.71_{\pm0.02}$ | $12.03_{\pm0.05}$ | $4.68_{\pm0.21}$ |
| AT-BSL | $37.78_{\pm0.24}$ | $35.46_{\pm0.12}$ | $24.13_{\pm0.47}$ | $10.98_{\pm0.38}$ | $10.16_{\pm0.32}$ | $3.75_{\pm0.66}$ | $39.49_{\pm0.05}$ | $37.16_{\pm0.19}$ | $26.00_{\pm1.25}$ | $12.11_{\pm0.15}$ | $11.28_{\pm0.25}$ | $4.41_{\pm0.52}$ |
| TAET | $36.83_{\pm0.25}$ | $33.35_{\pm0.40}$ | $20.22_{\pm0.80}$ | $14.02_{\pm0.17}$ | $12.33_{\pm0.28}$ | $4.45_{\pm0.45}$ | $36.77_{\pm0.61}$ | $33.10_{\pm0.66}$ | $19.83_{\pm0.34}$ | $14.05_{\pm0.09}$ | $12.11_{\pm0.14}$ | $4.55_{\pm0.55}$ |
| AT-SD | $35.84_{\pm0.05}$ | $33.89_{\pm0.05}$ | $20.89_{\pm0.25}$ | $\underline{16.30}_{\pm0.12}$ | $\underline{15.69}_{\pm0.17}$ | $\underline{6.07}_{\pm0.54}$ | $37.19_{\pm0.33}$ | $35.03_{\pm0.37}$ | $22.54_{\pm0.26}$ | $\underline{16.52}_{\pm0.10}$ | $\underline{15.93}_{\pm0.15}$ | $\underline{6.63}_{\pm0.66}$ |
| **PAT** | $\mathbf{41.16}_{\pm0.16}$ | $\mathbf{38.75}_{\pm0.27}$ | $\mathbf{25.23}_{\pm0.53}$ | $\mathbf{17.03}_{\pm0.27}$ | $\mathbf{16.06}_{\pm0.33}$ | $\mathbf{6.79}_{\pm0.12}$ | $\mathbf{43.26}_{\pm0.57}$ | $\mathbf{40.84}_{\pm0.66}$ | $\underline{26.81}_{\pm0.33}$ | $\mathbf{19.00}_{\pm0.17}$ | $\mathbf{18.05}_{\pm0.11}$ | $\mathbf{7.88}_{\pm0.59}$ |

*Table 4.* Worst-class accuracy and robustness against various $l_\infty$-attacks under different imbalance ratios (IRs) on CIFAR10-LT using ResNet-18. The **1st** and 2nd results are highlighted.

| IR | Method | Acc. | PGD | CW | BIM | AA |
|---|---|---|---|---|---|---|
| 10 | RoBal | $\mathbf{70.23}_{\pm1.13}$ | $29.36_{\pm0.47}$ | $26.70_{\pm0.44}$ | $29.03_{\pm0.45}$ | $23.83_{\pm0.38}$ |
| | REAT | $67.98_{\pm0.34}$ | $27.29_{\pm0.30}$ | $26.25_{\pm0.41}$ | $26.83_{\pm0.32}$ | $23.35_{\pm0.19}$ |
| | AT-BSL | $68.21_{\pm0.45}$ | $25.98_{\pm0.66}$ | $25.34_{\pm0.95}$ | $25.55_{\pm0.72}$ | $22.53_{\pm0.76}$ |
| | TAET | $66.07_{\pm0.53}$ | $\underline{33.27}_{\pm0.42}$ | $\underline{29.82}_{\pm0.40}$ | $\underline{32.96}_{\pm0.50}$ | $\underline{26.81}_{\pm1.18}$ |
| | AT-SD | $55.21_{\pm0.47}$ | $31.33_{\pm0.43}$ | $27.59_{\pm0.15}$ | $31.18_{\pm0.41}$ | $26.09_{\pm0.44}$ |
| | **PAT** | $\underline{70.15}_{\pm0.22}$ | $\mathbf{33.52}_{\pm0.37}$ | $\mathbf{31.00}_{\pm0.60}$ | $\mathbf{33.19}_{\pm0.48}$ | $\mathbf{28.08}_{\pm0.48}$ |
| 20 | RoBal | $\mathbf{66.39}_{\pm0.83}$ | $24.41_{\pm0.39}$ | $22.43_{\pm0.63}$ | $24.10_{\pm0.34}$ | $19.65_{\pm0.65}$ |
| | REAT | $64.17_{\pm1.07}$ | $23.51_{\pm0.67}$ | $22.46_{\pm0.75}$ | $23.09_{\pm0.69}$ | $19.83_{\pm0.50}$ |
| | AT-BSL | $64.51_{\pm0.35}$ | $20.19_{\pm0.72}$ | $19.79_{\pm0.96}$ | $19.89_{\pm0.92}$ | $17.38_{\pm0.79}$ |
| | TAET | $62.66_{\pm0.62}$ | $26.89_{\pm0.28}$ | $23.54_{\pm0.08}$ | $26.64_{\pm0.25}$ | $20.69_{\pm0.20}$ |
| | AT-SD | $53.79_{\pm0.87}$ | $\underline{28.75}_{\pm0.59}$ | $\underline{25.27}_{\pm0.60}$ | $\underline{28.59}_{\pm0.64}$ | $\underline{23.13}_{\pm0.22}$ |
| | **PAT** | $\underline{66.87}_{\pm0.35}$ | $\mathbf{29.69}_{\pm0.30}$ | $\mathbf{27.58}_{\pm0.37}$ | $\mathbf{29.37}_{\pm0.34}$ | $\mathbf{24.62}_{\pm0.27}$ |
| 50 | RoBal | $\underline{60.41}_{\pm0.19}$ | $19.36_{\pm1.48}$ | $16.93_{\pm1.10}$ | $19.17_{\pm1.48}$ | $14.73_{\pm1.03}$ |
| | REAT | $58.60_{\pm0.88}$ | $19.66_{\pm0.97}$ | $18.92_{\pm0.73}$ | $19.40_{\pm0.98}$ | $16.32_{\pm0.79}$ |
| | AT-BSL | $56.95_{\pm1.80}$ | $15.59_{\pm1.19}$ | $15.58_{\pm1.14}$ | $15.31_{\pm1.14}$ | $13.17_{\pm1.00}$ |
| | TAET | $49.73_{\pm5.45}$ | $17.79_{\pm0.94}$ | $13.93_{\pm0.77}$ | $17.63_{\pm0.89}$ | $11.75_{\pm0.96}$ |
| | AT-SD | $48.91_{\pm0.25}$ | $\underline{22.79}_{\pm0.29}$ | $\underline{20.36}_{\pm0.57}$ | $\underline{22.62}_{\pm0.33}$ | $\underline{18.28}_{\pm0.51}$ |
| | **PAT** | $\mathbf{64.03}_{\pm0.17}$ | $\mathbf{26.23}_{\pm0.42}$ | $\mathbf{23.27}_{\pm0.29}$ | $\mathbf{25.87}_{\pm0.37}$ | $\mathbf{20.81}_{\pm0.44}$ |
| 100 | RoBal | $53.01_{\pm1.04}$ | $14.85_{\pm1.08}$ | $12.39_{\pm0.82}$ | $14.61_{\pm1.20}$ | $10.65_{\pm0.91}$ |
| | REAT | $\underline{55.27}_{\pm0.71}$ | $\underline{17.29}_{\pm1.39}$ | $\underline{16.01}_{\pm1.36}$ | $\underline{17.01}_{\pm1.45}$ | $\underline{13.40}_{\pm1.14}$ |
| | AT-BSL | $51.60_{\pm1.56}$ | $12.28_{\pm0.63}$ | $11.95_{\pm0.53}$ | $11.93_{\pm0.61}$ | $9.80_{\pm0.47}$ |
| | TAET | $43.31_{\pm1.24}$ | $15.27_{\pm0.33}$ | $11.73_{\pm0.30}$ | $15.10_{\pm0.33}$ | $9.93_{\pm0.12}$ |
| | AT-SD | $42.77_{\pm1.60}$ | $15.95_{\pm0.91}$ | $14.23_{\pm1.32}$ | $15.86_{\pm1.13}$ | $12.39_{\pm0.94}$ |
| | **PAT** | $\mathbf{58.99}_{\pm1.06}$ | $\mathbf{21.06}_{\pm0.58}$ | $\mathbf{17.42}_{\pm0.39}$ | $\mathbf{20.79}_{\pm0.58}$ | $\mathbf{14.94}_{\pm0.26}$ |
| 200 | RoBal | $41.67_{\pm1.03}$ | $11.54_{\pm0.84}$ | $9.41_{\pm0.61}$ | $11.33_{\pm0.80}$ | $8.22_{\pm0.72}$ |
| | REAT | $\underline{48.09}_{\pm0.52}$ | $\underline{13.81}_{\pm1.05}$ | $\underline{12.52}_{\pm0.77}$ | $\underline{13.60}_{\pm1.03}$ | $\underline{10.18}_{\pm0.99}$ |
| | AT-BSL | $46.92_{\pm1.72}$ | $10.38_{\pm0.44}$ | $10.33_{\pm0.51}$ | $10.17_{\pm0.42}$ | $8.29_{\pm0.48}$ |
| | TAET | $34.84_{\pm3.49}$ | $12.53_{\pm0.50}$ | $9.84_{\pm0.28}$ | $12.45_{\pm0.51}$ | $8.19_{\pm0.29}$ |
| | AT-SD | $34.70_{\pm0.59}$ | $10.77_{\pm0.52}$ | $9.66_{\pm0.45}$ | $10.75_{\pm0.57}$ | $8.10_{\pm0.37}$ |
| | **PAT** | $\mathbf{49.01}_{\pm2.20}$ | $\mathbf{16.11}_{\pm0.26}$ | $\mathbf{13.56}_{\pm0.40}$ | $\mathbf{15.85}_{\pm0.28}$ | $\mathbf{11.30}_{\pm0.43}$ |

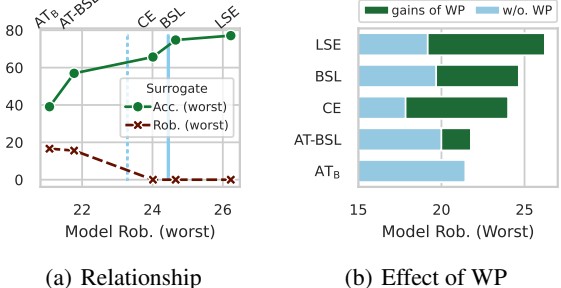

(a) Relationship  (b) Effect of WP

*Figure 3.* Relationship between posterior surrogate accuracy and resulting model robustness of worst-class, and the effect of Weight Perturbation (WP) on it. Blue lines in (a) correspond to using absolute labels without (dashed) and with (solid) label smoothing.

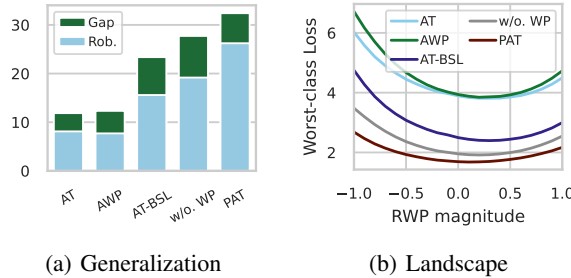

(a) Generalization  (b) Landscape

*Figure 4.* Relationship between weight loss landscape and robust generalization gap on worst-class. The landscape is visualized by Random Weight Perturbation (RWP) in (Wu et al., 2020). 'w/o. WP' denotes the variant of PAT that removes weight perturbation.

pronounced on tail- and worst-class, indicating that PAT effectively mitigates long-tail bias in adversarial training and yields a more balanced robustness profile. The consistent improvements across diverse datasets and architectures further demonstrate the robustness and general applicability of the proposed method.

We further evaluate PAT on CIFAR10-LT and CIFAR100-LT with imbalance ratios $\{10, 20, 50, 100, 200\}$ under multiple $l_\infty$ attacks beyond PGD, including AutoAttack (AA) (Croce & Hein, 2020), CW (Carlini & Wagner, 2017), and BIM (Kurakin et al., 2018), as reported in Tables 4 and 6. The results show that PAT consistently improves robustness across all imbalance ratios and attack settings, demonstrating stable performance under diverse and challenging conditions.

We observe that the relative performance gains gradually diminish under extremely severe imbalance regimes, where the approximation error term becomes large and dominant in the robust generalization bound. This behavior is consistent with our theoretical analysis and is further discussed in Appendix A.4.

### 6.2. Ablation studies

We conduct ablation studies to disentangle the contributions of the key components in PAT. Specifically, we evaluate four variants: (i) *w/o. Surrogate* removing the posterior surrogate and directly using absolute labels and class-frequency priors form the dataset; (ii) *w/o. Probability* replacing the probabilistic labels in Eq. (13) with their absolute counterparts;

*Table 5.* Accuracy, robustness, and training time of different variants on CIFAR10-LT and CIFAR100-LT using ResNet-18. Training time averaged per epoch across both datasets. The $1^{st}$ and $2^{nd}$ results are highlighted. WP denotes weight perturbation.

| Variant | Component | | | Effectiveness | | | | | | | | Efficiency | |
| --- | --- | --- | --- | --- | --- | --- | --- | --- | --- | --- | --- | --- | --- |
| | Posterior Surrogate | Fine-grained Label / Freq. | WP | CIFAR10-LT | | | | CIFAR100-LT | | | | Time (s) | |
| | | | | Acc. (all) | Acc. (worst) | Rob. (all) | Rob. (worst) | Acc. (all) | Acc. (worst) | Rob. (all) | Rob. (worst) | $h$ | $q$ |
| **PAT** | ✓ | ✓ / ✓ | ✓ | **73.41**$_{\pm0.55}$ | **64.03**$_{\pm0.17}$ | $\underline{36.71}_{\pm0.18}$ | **26.23**$_{\pm0.42}$ | **48.03**$_{\pm0.19}$ | 31.55$_{\pm0.27}$ | **22.11**$_{\pm0.19}$ | **9.27**$_{\pm0.39}$ | 19 | 4 |
| w/o. Surrogate | ✗ | ✗ / ✗ | ✓ | 70.93$_{\pm0.92}$ | 60.57$_{\pm1.25}$ | 35.59$_{\pm0.51}$ | 23.29$_{\pm1.07}$ | 45.52$_{\pm0.10}$ | 30.19$_{\pm0.64}$ | 19.50$_{\pm0.36}$ | 7.87$_{\pm0.43}$ | 19 | **0** |
| w/o. Probability | ✓ | ✗ / ✗ | ✓ | 56.00$_{\pm0.51}$ | 34.68$_{\pm0.99}$ | 31.45$_{\pm0.12}$ | 9.98$_{\pm0.21}$ | 43.34$_{\pm0.18}$ | 23.99$_{\pm0.77}$ | 19.59$_{\pm0.24}$ | 6.31$_{\pm0.28}$ | 19 | 4 |
| w/o. Freq. | ✓ | ✓ / ✗ | ✓ | 70.85$_{\pm0.59}$ | 59.47$_{\pm0.56}$ | **36.75**$_{\pm0.74}$ | $\underline{25.73}_{\pm0.46}$ | 47.36$_{\pm0.21}$ | **32.73**$_{\pm0.31}$ | $\underline{21.67}_{\pm0.11}$ | $\underline{9.24}_{\pm0.38}$ | 19 | 4 |
| w/o. WP | ✓ | ✓ / ✓ | ✗ | $\underline{72.10}_{\pm0.50}$ | $\underline{63.63}_{\pm0.49}$ | 31.66$_{\pm0.10}$ | 19.17$_{\pm0.55}$ | $\underline{47.55}_{\pm0.49}$ | $\underline{32.33}_{\pm0.77}$ | 19.56$_{\pm0.36}$ | 7.66$_{\pm0.34}$ | **16** | 4 |

*Figure 5.* Sensitivity of PAT to hyper-parameters $r \in [0.0, 1.0]$ and $\gamma \in [5 \times 10^{-4}, 5 \times 10^{-2}]$ on CIFAR10-LT using ResNet-18. Worst-class results are reported.

(a) mixing ratio $r$  (b) WP magnitude $\gamma$

(a) Comparision  (b) PAT  (c) AT-BSL

*Figure 6.* Worst-class accuracy and robustness under data augmentations including RandAugment (RA) (Cubuk et al., 2020) and AutoAugment (AuA) (Cubuk et al., 2019) using ResNet-18 on CIFAR10-LT.

(iii) *w/o. Freq.* replacing the fine-grained frequency update in Eq. (14) with the dataset class-frequency prior; and (iv) *w/o. WP* removing the weight perturbation in Eq. (16). All other components are kept identical to PAT. Results on CIFAR10-LT and CIFAR100-LT are reported in Table 5, with additional results on TinyImageNet-LT provided in Table 7 in the appendix.

**Ablation of posterior surrogate.** Compared with *w/o. Surrogate*, PAT consistently achieves higher accuracy and robustness, validating the benefit of approximating Bayesian labels via a posterior surrogate. To further examine our choice of training the surrogate with the LSE objective (Eq. (11)), we compare it with alternative objectives in Figs. 3 and 9. Specifically, we consider surrogates trained with: (i) standard objectives, including cross-entropy (CE), Balanced Softmax Loss (BSL), and LSE; (ii) adversarial objectives, including AT-BSL and $\text{AT}_\text{B}$ (the balanced teacher used in AT-SD). We also include label smoothing, which provides data-independent probabilistic labels, as a baseline. The results align with our theory: accurate posterior approximation is crucial for long-tailed robustness. While label smoothing improves over absolute labels, surrogates trained with BSL or LSE achieve stronger robustness, highlighting the necessity of data-dependent posterior surrogates.

**Ablation of fine-grained label and frequency.** Variants *w/o. Surrogate* and *w/o. Probability* both suffer notable performance degradation, confirming the importance of fine-grained probabilistic supervision. Moreover, compared with *w/o. Freq.*, PAT yields clear improvements in worst-class robustness, demonstrating the benefit of estimating class frequencies from posterior-aware labels. An intuitive visualization in Fig. 8 (in appendix) further shows that posterior mismatch can lead to overestimation of class imbalance, which in turn harms robust generalization.

**Ablation of weight perturbation (WP).** PAT consistently outperforms *w/o. WP*, particularly on worst-class robustness. As shown in Figs. 4 and 10, a smaller robust generalization gap consistently coincides with a flatter weight loss landscape, confirming the complementary role of WP. We further analyze the interaction between posterior approximation quality and WP in Figs. 3 and 9. The results show that only with WP does improved surrogate accuracy reliably translate into stronger robustness of the target model, consistent with our theoretical analysis. Notably, Figs. 4 and 10 also show that AWP—applying WP to imbalance-unaware AT—can exhibit a larger robust generalization gap than AT in long-tailed settings, underscoring the necessity of combining posterior-aware supervision with WP for better long-tailed robustness.

### 6.3. Sensitivity experiments

**Hyper-parameters.** We evaluate the sensitivity of PAT to the hyper-parameters in Fig. 5, and Table 8 and Fig. 11 in the appendix. We have the following observations.

- *LSE temperature $\tau$.* Table 8 in appendix shows that the optimal $\tau$ scales proportionally with the number of classes $C$, which yields a tight and smooth approximation to the max operator, and thus we simply suggest $\tau = 10^{\lceil \log C \rceil - 1}$.
- *Mixing ratio $r$.* As shown in Figs. 5(a) and 11(a), accuracy generally increases with $r$, as the objective increasingly aligns the prediction behaviors of the target model with that of the surrogate. This resembles distillation and transfers the surrogate's standard generalization. Impor-

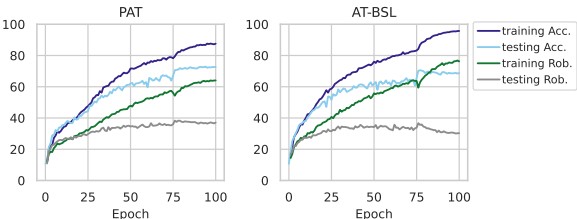

*Figure 7.* Learning curves of PAT and AT-BSL on CIFAR10-LT with ResNet-18. The clean and robust accuracies (denoted as Acc. and Rob.) averaged over all classes are reported.

tantly, robustness typically peaks at an intermediate value within $r$. Moderate $r$ helps capture local posterior structure around adversarial data that helps mitigate posterior mismatch. When $r$ is large (e.g., $r = 1.0$), the fine-grained supervision is dominated by adversarial soft label, weakening consistency between clean and adversarial data and harming robustness. We set $r = 0.6$ for CIFAR10-LT, and $r = 0.8$ for CIFAR100-LT and TinyImageNet-LT, selected based on validation robustness and observed to generalize well across architectures.

- *Weight perturbation strength* $\gamma$. In Figs. 5(b) and 11(b), moderate $\gamma$ improve robustness, while excessively large ones hurt accuracy and robustness. We therefore use $\gamma = 5 \times 10^{-3}$, which is stable across datasets and architectures.

Other hyper-parameters (e.g., learning rate $\eta = 0.1$) follow standard practices. Overall, PAT does not require extensive tuning: most parameters are either fixed by design (e.g., $\tau$) or robust across different settings (e.g., $\gamma$) and only $r$ may benefit from light tuning. In practice, $r = 0.6$ can serve as a robust default when hyper-parameter tuning is limited.

**Data augmentations.** We evaluate PAT under different data augmentations. As shown in Fig. 6, PAT consistently outperforms AT-BSL. While augmentation remains complementary, the marginal gain from it is slightly smaller for PAT than AT-BSL, consistent to our analysis in Remark 4.10.

**Training time.** Training time per epoch results in Table 5 and Table 7 shows that surrogate and weight perturbations introduce a modest computational overhead, which we consider acceptable given the substantial gains in robustness.

**Training stability.** The learning curves in Fig. 7 show that PAT trains stably without noticeable oscillations or drift. Moreover, compared with AT-BSL, PAT exhibits smaller generalization gaps, i.e., smaller differences between training and testing Acc./Rob.

## 7. Conclusion

Adversarial training under long-tailed distributions fails most severely on the worst-class (often tail). In this work, we show that this failure stems from a posterior mismatch caused by coarse-grained absolute labeling, which distorts

frequency estimation, enlarges the robust generalization gap, and ultimately worsen the long-tailed adversarial robustness. We propose PAT, a posterior-driven adversarial training framework that combines fine-grained supervision with landscape-aware optimization via weight perturbations. Across diverse long-tailed benchmarks, PAT delivers consistent robustness improvements, especially on worst-class.

## Acknowledgements

This work was supported by the Science and Technology Major Project of Sichuan Province (2024ZDZX0003, 2025ZDZX0140), and the National Key R&D Program of China (2024YFB3312503). We also acknowledge the support of Sichuan Province Engineering Technology Research Center of Broadband Electronics Intelligent Manufacturing.

## Impact statement

This work studies long-tailed adversarial robustness and shows that robustness degradation on tail/worst-class is fundamentally driven by posterior mismatch induced by coarse-grained absolute labels. By identifying posterior estimation as a key factor in long-tailed robust generalization, our analysis highlights the need to jointly consider robustness, class imbalance, and generalization.

Beyond robustness, this perspective may inform other imbalance-sensitive problems such as fairness-aware learning, domain shift, and rare-event prediction. As with prior robustness research, misuse of these techniques could enable more evasive or manipulative systems. We encourage responsible deployment practices, including transparency and auditing, to mitigate such risks.

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

# A. Discussions

## A.1. Relation to fairness-oriented adversarial training

Prior work has shown that even under class-balanced data distributions, adversarial training can induce substantial class-wise robustness disparaties, with certain classes being consistently less robust. To mitigate such disparities, a line of research focuses on improving fairness in adversarial robustness via reweighting schemes, adaptive margins, or class-aware objectives (Xu et al., 2021; Ma et al., 2022; Sun et al., 2023; Wei et al., 2023; Li & Liu, 2023; Zhang et al., 2024; Zhao et al., 2024; Lee et al., 2024).

These approaches, however, typically assume balanced training data and primarily target intrinsic class difficulty. In contrast, real-world data are often long-tailed, where class imbalance not only amplifies robustness disparities but also introduces bias in frequency estimation and generalization. As a result, robustness degradation in long-tailed settings cannot be fully explained or addressed by intrinsic class difficulty alone.

## A.2. Relation to distillation techniques

Although PAT bears superficial resemblance to Knowledge Distillation (KD) (Hinton et al., 2014) and Adversarial Robustness Distillation (ARD) (Goldblum et al., 2020), its motivation and role are fundamentally different.

Conceptually, as illustrated in Fig. 2, KD aims to transfer generalization ability from a large teacher to a smaller student, while ARD extends this paradigm by distilling adversarial robustness from a robust teacher. In both cases, the teacher serves as a source of performance transfer. In contrast, we introduce a theoretically motivated posterior surrogate whose purpose is not to transfer either standard or robust generalization, but to correct posterior mismatch induced by coarse-grained absolute labels.

Practically, PAT remains a bona fide adversarial training method, as adversarial examples are explicitly incorporated into the optimization of the target model. This distinguishes it fundamentally from standard KD approaches. Moreover, the posterior surrogate in PAT is trained solely on clean data, incurring substantially lower computational cost than adversarially training a robust teacher, as required by ARD-based methods such as AT-SD (Cho et al., 2025). Importantly, robustness of the surrogate itself is not required in our framework as its role is to approximate class posteriors, a property that is empirically validated in Figs. 3 and 9.

## A.3. Scope of the theory

Theorems 4.5 and 4.9 are derived under the assumption that $\mathcal{L}_{\text{BSL}}(h, p) \geq \mathcal{R}(h, p)$, i.e., minimizing the population BSL objective leads to improved robust generalization on $p$.

This assumption is well motivated. Specifically, Lemma 3.1 shows that the population BSL objective encourages the model to capture mutual information between adversarial inputs $\tilde{x}$ and labels $y$, which is essential for learning $p(y|\tilde{x})$ and reducing the robust risk $\mathcal{R}(h, p)$.

More broadly, our analysis is not restricted to BSL and extends to other objectives, provided they guide the model toward robust generalization on distribution $p$.

## A.4. Limitations

PAT does not assume access to Bayesian labels or a perfectly accurate class posterior estimator. Bayesian labels are introduced solely as a theoretical reference to formalize posterior mismatch, and our bounds in Theorems 4.7 and 4.9 explicitly characterize how approximation error propagates into class-frequency estimation and robust generalization.

In practice, the effectiveness of PAT depends on whether the available data support learning a sufficiently accurate posterior surrogate. Under extremely scarce data or severely imbalanced regimes, posterior approximation may become unreliable, diminishing robustness gains, as observed in Table 6. Notably, such regimes often coincide with very low clean accuracy, where improving robustness against attacks may lose its practical significance.

# B. Theoretical proofs

## B.1. Useful inequality

**Lemma B.1** (Bennett's inequality cf. Theorem 3 of (Maurer & Pontil, 2009)). *Let $z$ be a random variables with values in $[0, \Delta]$ follows a distribution $p$, set $S = \{z^{(k)}\}_{k \in [N]} \sim p^N$ and $\delta \in (0, 1)$, we have with probability at least $1 - \delta$ that*

$$\mathbb{E}_p[z] - \mathbb{E}_S[z] \leq \sqrt{\frac{2\text{Var}[z]\log(1/\delta)}{N}} + \Delta\frac{\log(1/\delta)}{3N}.$$

## B.2. Nature of BSL

*Proof of Lemma 3.1.* When $\tau_b = 1$, the balanced softmax in Eq. (6) can be rewritten as:

$$\tilde{\mathbf{y}}_h^{\text{BSL}} = \left[\frac{\exp(h(\tilde{x})_i + \log p(y_i))}{\sum_{y_j \in \mathcal{Y}} \exp(h(\tilde{x})_j + \log p(y_j))}\right]_{i \in [C]}$$

$$= \left[\frac{p(y_i)\exp(h(\tilde{x})_i)}{\mathbb{E}_{(\cdot, y_j) \sim p}[\exp(h(\tilde{x})_j)]}\right]_{i \in [C]}.$$

Let $p'(x, y)$ be an arbitrary distribution over instance space $\mathcal{X} \times \mathcal{Y}$. According to the definition of mutual information,

$$\text{MI}(x, y) = \mathbb{E}_{(x,y) \sim p}\left[\log\frac{p(x, y)}{p(x)p(y)}\right]$$

$$= D_{\text{KL}}(p, p') + \mathbb{E}_{(x,y) \sim p}\left[\log\frac{p'(x, y)}{p(x)p(y)}\right].$$

Following (McAllester & Stratos, 2020), we parameterize $p'$ as $p'(x, y_i) = \frac{\exp(h(x)_i)}{\mathbb{E}_{(\cdot, y_j) \sim p}[\exp(h(\tilde{x})_j)]} p(x)p(y)$. Substituting this form yields,

$$
\begin{aligned}
\mathrm{MI}(x, y) &\geq \mathbb{E}_{(x, y_i) \sim p}\left[ \log \frac{\exp(h(x)_i)}{\mathbb{E}_{(\cdot, y_j) \sim p}[\exp(h(\tilde{x})_j)]} \right] \\
&= \mathbb{E}_{(x, y_i) \sim p}\left[ \log \frac{p(y_i) \exp(h(x)_i)}{\mathbb{E}_{(\cdot, y_j) \sim p}[\exp(h(x)_j)]} \right] + \mathrm{H}(y).
\end{aligned}
$$

Now consider adversarial examples generated by a measurable transport map that perturbs each $(x, y) \sim p$ to $(\tilde{x}, y)$. This induces an adversarial distribution $\tilde{p}$ over $(\tilde{x}, y)$. Applying the same argument under $\tilde{p}$ yields

$$
\begin{aligned}
\mathrm{MI}(\tilde{x}, y) &\geq \mathbb{E}_{(\tilde{x}, y_i) \sim \tilde{p}}\left[ \log \frac{p(y_i) \exp(h(\tilde{x})_i)}{\mathbb{E}_{(\cdot, y_j) \sim p}[\exp(h(\tilde{x})_j)]} \right] + \mathrm{H}(y) \\
&= -\mathcal{L}_{\mathrm{BSL}}(h, p) + \mathrm{H}(y).
\end{aligned}
$$

Rearranging terms gives $\mathcal{L}_{\mathrm{BSL}}(h, p) \geq \mathrm{H}(y) - \mathrm{MI}(x, y)$. $\square$

## B.3. Effect 1: Biased class-frequency estimation

*Proof of Lemma 4.1.* By the definition of variance,

$$
\mathrm{Var}[\mathbf{y}_i] = \mathbb{E}_{(x, \cdot) \sim p}\left[\mathbf{y}_i^2\right] - \left(\mathbb{E}_{(x, \cdot) \sim p}[\mathbf{y}_i]\right)^2,
$$

and

$$
\begin{aligned}
\mathrm{Var}[\mathbf{e}_{y,i}] &= \mathbb{E}_{(x, y) \sim p}\left[\mathbf{e}_{y,i}^2\right] - \left(\mathbb{E}_{(x, y) \sim p}[\mathbf{e}_{y,i}]\right)^2 \\
&= \mathbb{E}_{(x, y) \sim p}[\mathbf{e}_{y,i}] - \left(\mathbb{E}_{(x, y) \sim p}[\mathbf{e}_{y,i}]\right)^2 \\
&= \mathbb{E}_{(x, \cdot) \sim p}[\mathbf{y}_i] - \left(\mathbb{E}_{(x, \cdot) \sim p}[\mathbf{y}_i]\right)^2,
\end{aligned}
$$

where the last two lines hold because $\mathbf{e}_{y,i}^2 = \mathbf{e}_{y,i}$ and $\mathbb{E}_{y|x \sim p}[\mathbf{e}_y] = \mathbf{y}$ ($\mathbf{e}_y$ is an unbiased estimation of $\mathbf{y}$).

Now we get the difference between the two variance as:

$$
\mathrm{Var}[\mathbf{y}_i] - \mathrm{Var}[\mathbf{e}_{y,i}] = \mathbb{E}_{(x, \cdot) \sim p}\left[\mathbf{y}_i^2 - \mathbf{y}_i\right] \leq 0.
$$

Therefore, $\mathrm{Var}[\mathbf{y}_i] \leq \mathrm{Var}[\mathbf{e}_{y,i}]$, and the equality holds only when $\mathbf{y}_i = \mathbf{e}_{y,i}$ over $p$. $\square$

*Proof of Theorem 4.2.* (i) According to Bennett's inequality (Lemma B.1), for $\forall y_i \in \mathcal{Y}$,

$$
\begin{aligned}
&\left| \mathbb{E}_{(x, \cdot) \in S}[\mathbf{y}_i] - p(y_i) \right| \\
&= \left| \mathbb{E}_{(x, \cdot) \in S}[\mathbf{y}_i] - \mathbb{E}_{S \sim p^N} \mathbb{E}_{(x, \cdot) \in S}[\mathbf{y}_i] \right| \\
&\leq \sqrt{\frac{2\mathrm{Var}[\mathbf{y}_i] \log(2C/\delta)}{N}} + \frac{\log(2C/\delta)}{3N} \\
&= \sqrt{\mathrm{Var}[\mathbf{y}_i]} A_1 + A_2,
\end{aligned}
$$

where the inequality holds with a probability at least $1 - \delta$.

(ii) Since $\mathbb{E}_{(x, y) \sim p}[\mathbf{e}_{y,i}] = \mathbb{E}_{(x, \cdot) \sim p}[\mathbf{y}_i] = p(y_i)$, we can similarly get that for $\forall y_i \in \mathcal{Y}$:

$$
\begin{aligned}
&\left| \mathbb{E}_{(x, y) \in S}[\mathbf{e}_{y,i}] - p(y_i) \right| \\
&= \left| \mathbb{E}_{(x, y) \in S}[\mathbf{e}_{y,i}] - \mathbb{E}_{S \sim p^N} \mathbb{E}_{(x, \cdot) \in S}[\mathbf{e}_{y,i}] \right| \\
&\leq \sqrt{\frac{2\mathrm{Var}[\mathbf{e}_{y,i}] \log(2C/\delta)}{N}} + \frac{\log(2C/\delta)}{3N} \\
&= \sqrt{\mathrm{Var}[\mathbf{e}_{y,i}]} A_1 + A_2
\end{aligned}
$$

holds with a probability at least $1 - \delta$. $\square$

## B.4. Effect2: Enlarged robust generalization gap

*Proof of Lemma 4.4.* By the definition of variance,

$$
\begin{aligned}
&\mathrm{Var}[\ell(\mathbf{y}, \tilde{\mathbf{y}}_h^{\mathrm{BSL}})] \\
&= \mathbb{E}_{(x, \cdot) \sim p}\left[(-\mathbf{y}^\top \log \tilde{\mathbf{y}}_h^{\mathrm{BSL}})^2\right] - \left(\mathbb{E}_{(x, \cdot) \sim p}[-\mathbf{y}^\top \log \tilde{\mathbf{y}}_h^{\mathrm{BSL}}]\right)^2,
\end{aligned}
$$

and since $\mathbb{E}_{y|x \sim p}[\mathbf{e}_y] = \mathbf{y}$,

$$
\begin{aligned}
&\mathrm{Var}[\ell(\mathbf{e}_y, \tilde{\mathbf{y}}_h^{\mathrm{BSL}})] \\
&= \mathbb{E}_{(x, y) \sim p}\left[(-\mathbf{e}_y^\top \log \tilde{\mathbf{y}}_h^{\mathrm{BSL}})^2\right] - \left(\mathbb{E}_{(x, y) \sim p}[-\mathbf{e}_y^\top \log \tilde{\mathbf{y}}_h^{\mathrm{BSL}}]\right)^2 \\
&= \mathbb{E}_{(x, y) \sim p}\left[\mathbf{e}_y^\top(-\log \tilde{\mathbf{y}}_h^{\mathrm{BSL}})^2\right] - \left(\mathbb{E}_{(x, y) \sim p}[-\mathbf{e}_y^\top \log \tilde{\mathbf{y}}_h^{\mathrm{BSL}}]\right)^2 \\
&= \mathbb{E}_{(x, \cdot) \sim p}\left[\mathbf{y}^\top(-\log \tilde{\mathbf{y}}_h^{\mathrm{BSL}})^2\right] - \left(\mathbb{E}_{(x, \cdot) \sim p}[-\mathbf{y}^\top \log \tilde{\mathbf{y}}_h^{\mathrm{BSL}}]\right)^2.
\end{aligned}
$$

Then, we get the difference between the above two as:

$$
\begin{aligned}
&\mathrm{Var}[\ell(\mathbf{y}, \tilde{\mathbf{y}}_h^{\mathrm{BSL}})] - \mathrm{Var}[\ell(\mathbf{e}_y, \tilde{\mathbf{y}}_h^{\mathrm{BSL}})] \\
&= \mathbb{E}_{(x, \cdot) \sim p}\left[(-\mathbf{y}^\top \log \tilde{\mathbf{y}}_h^{\mathrm{BSL}})^2\right] - \mathbb{E}_{(x, \cdot) \sim p}\left[\mathbf{y}^\top(-\log \tilde{\mathbf{y}}_h^{\mathrm{BSL}})^2\right] \\
&= \mathbb{E}_{(x, \cdot) \sim p}\left[\left(\mathbb{E}_{y|x}[-\mathbf{e}_y^\top \log \tilde{\mathbf{y}}_h^{\mathrm{BSL}}]\right)^2 - \mathbb{E}_{y|x}[(-\mathbf{e}_y^\top \log \tilde{\mathbf{y}}_h^{\mathrm{BSL}})^2]\right] \\
&= -\mathbb{E}_{(x, \cdot) \sim p}\left[\mathrm{Var}_{y|x}[\ell(\mathbf{e}_y, \tilde{\mathbf{y}}_h^{\mathrm{BSL}})]\right] \leq 0.
\end{aligned}
$$

Therefore, $\mathrm{Var}[\ell(\mathbf{y}, \tilde{\mathbf{y}}_h^{\mathrm{BSL}})] \leq \mathrm{Var}[\ell(\mathbf{e}_y, \tilde{\mathbf{y}}_h^{\mathrm{BSL}})]$, and the equality holds when $\ell(\mathbf{e}_y, \tilde{\mathbf{y}}_h^{\mathrm{BSL}})$ is a constant over $p$. $\square$

*Proof of Theorem 4.5.* By Lemma B.1, with probability at least $1 - \delta$ over the draw of the sample $S$, the following concentration inequality holds uniformly for all $u \in \mathcal{U}$:

$$
\begin{aligned}
&\mathcal{L}_{\mathrm{BSL}}(h + u, p) - \mathcal{L}_{\mathrm{BSL}}(h + u, S) \\
&= \mathbb{E}_{(x, y) \sim p}[\ell(\mathbf{e}_y, \tilde{\mathbf{y}}_{h+u}^{\mathrm{BSL}})] - \mathbb{E}_{(x, y) \in S}[\ell(\mathbf{e}_y, \tilde{\mathbf{y}}_{h+u}^{\mathrm{BSL}})] \\
&\leq \sqrt{\frac{2\mathrm{Var}[\ell(\mathbf{e}_y, \tilde{\mathbf{y}}_{h+u}^{\mathrm{BSL}})] \log(|\mathcal{U}|/\delta)}{N}} + \Delta_{h+u} \frac{\log(|\mathcal{U}|/\delta)}{3N} \\
&= \sqrt{\mathrm{Var}[\ell(\mathbf{e}_y, \tilde{\mathbf{y}}_{h+u}^{\mathrm{BSL}})]} C_1 + C_2.
\end{aligned}
$$

Moreover, by Lemma 3.1, the population BSL objective upper-bounds the robust error, i.e., $\mathcal{L}_{\mathrm{BSL}}(h, p) \geq \mathcal{R}(h, p)$. Combining the above inequalities and taking expectation over $u \in \mathcal{U}$ yields, with probability at least $1 - \delta$,

$$
\begin{aligned}
&\mathbb{E}_{u \in \mathcal{U}}\left[\mathcal{R}(h + u, p) - \mathcal{L}_{\mathrm{BSL}}(h + u, S)\right] \\
&\leq \mathbb{E}_{u \in \mathcal{U}}\left[\sqrt{\mathrm{Var}[\ell(\mathbf{e}_y, \tilde{\mathbf{y}}_{h+u}^{\mathrm{BSL}})]} C_1 + C_2\right],
\end{aligned}
$$

Finally, we decompose the upper bound of the gap as:

$$\mathbb{E}_{u\in\mathcal{U}}\big[\mathcal{R}(h+u,p)\big]-\mathcal{L}_{\mathrm{BSL}}(h,S)$$

$$=\mathbb{E}_{u\in\mathcal{U}}\big[\mathcal{R}(h+u,p)-\mathcal{L}_{\mathrm{BSL}}(h+u,S)$$
$$+\mathcal{L}_{\mathrm{BSL}}(h+u,S)-\mathcal{L}_{\mathrm{BSL}}(h,S)\big]$$

$$\leq\mathbb{E}_{u\in\mathcal{U}}\Big[\sqrt{\mathrm{Var}[\ell(\mathbf{e}_y,\tilde{\mathbf{y}}_{h+u}^{\mathrm{BSL}})]}\,C_1+C_2\Big]$$
$$+\mathbb{E}_{u\in\mathcal{U}}\big[\mathcal{L}_{\mathrm{BSL}}(h+u,S)-\mathcal{L}_{\mathrm{BSL}}(h,S)\big],$$

with a probability at least $1-\delta$. $\qquad\square$

### B.5. Guideline 1: A surrogate accurate even on worst-class

*Proof of Theorem 4.7.* According to Bennett's inequality (Lemma B.1), for $\forall y_i\in\mathcal{Y}$,

$$\big|\mathbb{E}_{(x,\cdot)\in S}[\mathbf{y}_{q,i}]-\mathbb{E}_{(x,\cdot)\sim p}[\mathbf{y}_{q,i}]\big|$$
$$\leq\sqrt{\frac{2\mathrm{Var}[\mathbf{y}_{q,i}]\log(2C/\delta)}{N}}+\frac{\log(2C/\delta)}{3N}$$

holds with a probability at least $1-\delta$. Meanwhile,

$$\big|\mathbb{E}_{(x,\cdot)\sim p}[\mathbf{y}_{q,i}]-\mathbb{E}_{(x,\cdot)\sim p}[\mathbf{y}_i]\big|\leq\mathbb{E}_{(x,\cdot)\sim p}\big[|\mathbf{y}_{q,i}-\mathbf{y}_i|\big],$$

where $\mathbf{y}_{q,i}$ and $\mathbf{y}_i$ are the $i$-th dimension of $\mathbf{y}_q$ and $\mathbf{y}$, respectively. Therefore,

$$\big|\mathbb{E}_{(x,\cdot)\in S}[\mathbf{y}_{q,i}]-p(y_i)\big|\leq\mathbb{E}_{(x,\cdot)\sim p}[|\mathbf{y}_{q,i}-\mathbf{y}_i|]$$
$$+\sqrt{\frac{2\mathrm{Var}[\mathbf{y}_{q,i}]\log(2C/\delta)}{N}}+\frac{\log(2C/\delta)}{3N},$$

holds with a probability at least $1-\delta$ for $\forall y_i\in\mathcal{Y}$. $\qquad\square$

### B.6. Guideline 2: A flat weight loss landscape.

*Proof of Theorem 4.9.* By Lemma B.1, with probability at least $1-\delta$ over the draw of the sample $S$, the following concentration inequality holds uniformly for all $u\in\mathcal{U}$:

$$\mathcal{L}_{\mathrm{BSL}}(h+u,q)-\mathcal{L}_{\mathrm{BSL}}(h+u,S_q)$$
$$=\mathbb{E}_{(x,\cdot)\sim p}[\ell(\mathbf{y}_q,\tilde{\mathbf{y}}_{h+u}^{\mathrm{BSL}})]-\mathbb{E}_{(x,\cdot)\in S}[\ell(\mathbf{y}_q,\tilde{\mathbf{y}}_{h+u}^{\mathrm{BSL}})]$$
$$\leq\sqrt{\frac{2\mathrm{Var}[\ell(\mathbf{y}_q,\tilde{\mathbf{y}}_{h+u}^{\mathrm{BSL}})]\log(|\mathcal{U}|/\delta)}{N}}+\Delta_{h+u}\frac{\log(|\mathcal{U}|/\delta)}{3N}$$
$$=\sqrt{\mathrm{Var}[\ell(\mathbf{y}_q,\tilde{\mathbf{y}}_{h+u}^{\mathrm{BSL}})]}\,C_1+C_2.$$

Next, for any hypothesis $h\in\mathcal{H}$ and weight perturbation $u\in\mathcal{U}$, we first note that

$$\mathbb{E}_{u\in\mathcal{U}}\big[\mathcal{R}(h+u,p)-\mathcal{R}(h+u,q)\big]$$
$$\leq\mathbb{E}_{u\in\mathcal{U}}\big[\sum_{i\in[C]}\mathbb{E}_{(x,\cdot)\sim p}[|\mathbf{y}_i-\mathbf{y}_{q,i}|\log\tilde{\mathbf{y}}_{h+u,i}]\big]$$
$$\leq\mathbb{E}_{u\in\mathcal{U}}\big[\sum_{i\in[C]}\mathbb{E}_{(x,\cdot)\sim p}[|\mathbf{y}_i-\mathbf{y}_{q,i}|]\max_{x\in\mathcal{X},i\in[C]}\{-\log\tilde{\mathbf{y}}_{h+u,i}\}\big]$$
$$=\mathbb{E}_{u\in\mathcal{U}}[\Delta_{h+u}]\sum_{i\in[C]}\mathbb{E}_{(x,\cdot)\sim p}[|\mathbf{y}_i-\mathbf{y}_{q,i}|].$$

Moreover, by Lemma 3.1, $\mathcal{L}_{\mathrm{BSL}}(h,q)\geq\mathcal{R}(h,q)$. Combining the above inequalities and taking expectation over $u\in\mathcal{U}$ yields, with probability at least $1-\delta$,

$$\mathbb{E}_{u\in\mathcal{U}}\big[\mathcal{R}(h+u,p)-\mathcal{L}_{\mathrm{BSL}}(h+u,S_q)\big]$$
$$\leq\mathbb{E}_{u\in\mathcal{U}}[\Delta_{h+u}]\sum_{i\in[C]}\mathbb{E}_{(x,\cdot)\sim p}[|\mathbf{y}_{q,i}-\mathbf{y}_i|]$$
$$+\mathbb{E}_{u\in\mathcal{U}}\Big[\sqrt{\mathrm{Var}[\ell(\mathbf{y}_q,\tilde{\mathbf{y}}_{h+u}^{\mathrm{BSL}})]}\,C_1+C_2\Big].$$

Finally, we decompose the upper bound of excess gap as

$$\mathbb{E}_{u\in\mathcal{U}}\big[\mathcal{R}(h+u,p)\big]-\mathcal{L}_{\mathrm{BSL}}(h,S_q)$$
$$=\mathbb{E}_{u\in\mathcal{U}}\big[\mathcal{R}(h+u,p)-\mathcal{L}_{\mathrm{BSL}}(h+u,S_q)$$
$$+\mathcal{L}_{\mathrm{BSL}}(h+u,S_q)\big]-\mathcal{L}_{\mathrm{BSL}}(h,S_q)$$
$$\leq\mathbb{E}_{u\in\mathcal{U}}\Big[\sqrt{\mathrm{Var}[\ell(\mathbf{y}_q,\tilde{\mathbf{y}}_{h+u}^{\mathrm{BSL}})]}\,C_1+C_2\Big]$$
$$+\mathbb{E}_{u\in\mathcal{U}}\big[\mathcal{L}_{\mathrm{BSL}}(h+u,S_q)-\mathcal{L}_{\mathrm{BSL}}(h,S_q)\big]$$
$$+\mathbb{E}_{u\in\mathcal{U}}[\Delta_{h+u}]\sum_{i\in[C]}\mathbb{E}_{(x,\cdot)\sim p}[|\mathbf{y}_{q,i}-\mathbf{y}_i|]$$

with a probability at least $1-\delta$. $\qquad\square$

## C. Reproduction

We provide PyTorch-style pseudocode in Algorithm 1 to facilitate understanding of PAT, and our code repository is publicly available online[5].

## D. Additional experiments

### D.1. Detailed configurations

For PAT, the posterior surrogate is trained using SGD with Nesterov momentum (Nesterov, 1983), with a learning rate of 0.1, weight decay $5\times10^{-4}$, and momentum 0.9. We use a batch size of 128 and train the surrogate for 200 epochs, decaying the learning rate by a factor of 5 at the 60th, 120th, and 160th epochs. The target model is trained adversarially for 100 epochs with the same batch size. The learning rate is decayed by a factor of 10 at the 75th and 90th epochs, following the protocol in (Zhang et al., 2019). Adversarial examples are generated using PGD (Madry et al., 2018) with 10 iterations, step size $\alpha=2/255$, and budget $\epsilon=8/255$.

Following (Pang et al., 2021), all adversarial training experiments use ReLU activations and set batch normalization layers to evaluation mode. The target model is optimized using SGD with Nesterov momentum, with the same learning rate, weight decay, and momentum as the surrogate. Hyperparameters of all baseline methods are set to the optimal values reported in their respective papers. All experiments are conducted on a single NVIDIA RTX 4090 GPU.

---

[5]https://github.com/zhang-lilin/long-tailed-robustness

*Table 6.* Worst-class accuracy and robustness against various $l_\infty$-attacks under different imbalance ratios (IRs) on CIFAR100-LT using ResNet-18. The **1st** and 2nd results are highlighted.

| IR | Method | Acc. | PGD | CW | BIM | AA |
|---|---|---|---|---|---|---|
| 10 | RoBal | $\underline{30.49}_{\pm 0.76}$ | $7.22_{\pm 0.07}$ | $5.89_{\pm 0.41}$ | $7.07_{\pm 0.48}$ | $4.32_{\pm 0.41}$ |
| | REAT | $30.47_{\pm 0.46}$ | $6.49_{\pm 0.37}$ | $5.69_{\pm 0.11}$ | $6.37_{\pm 0.18}$ | $4.58_{\pm 0.14}$ |
| | AT-BSL | $29.51_{\pm 0.50}$ | $5.91_{\pm 0.10}$ | $5.15_{\pm 0.17}$ | $5.72_{\pm 0.04}$ | $4.23_{\pm 0.25}$ |
| | TAET | $27.21_{\pm 0.48}$ | $6.94_{\pm 0.12}$ | $4.87_{\pm 0.25}$ | $6.87_{\pm 0.15}$ | $3.87_{\pm 0.17}$ |
| | AT-SD | $25.12_{\pm 0.59}$ | $\mathbf{9.15}_{\pm 0.22}$ | $\underline{6.85}_{\pm 0.16}$ | $\underline{9.07}_{\pm 0.46}$ | $\mathbf{5.88}_{\pm 0.20}$ |
| | **PAT** | $\mathbf{31.55}_{\pm 0.27}$ | $\underline{9.27}_{\pm 0.39}$ | $\mathbf{7.00}_{\pm 0.19}$ | $\mathbf{9.15}_{\pm 0.25}$ | $\underline{5.63}_{\pm 0.12}$ |
| 20 | RoBal | $\underline{26.29}_{\pm 0.51}$ | $5.54_{\pm 0.15}$ | $4.40_{\pm 0.20}$ | $5.47_{\pm 0.34}$ | $3.22_{\pm 0.13}$ |
| | REAT | $26.11_{\pm 0.43}$ | $4.91_{\pm 0.58}$ | $4.09_{\pm 0.07}$ | $4.77_{\pm 0.45}$ | $3.29_{\pm 0.34}$ |
| | AT-BSL | $25.55_{\pm 0.06}$ | $4.28_{\pm 0.26}$ | $3.69_{\pm 0.31}$ | $4.13_{\pm 0.15}$ | $2.99_{\pm 0.20}$ |
| | TAET | $20.58_{\pm 0.38}$ | $4.28_{\pm 0.15}$ | $3.09_{\pm 0.37}$ | $4.21_{\pm 0.26}$ | $2.42_{\pm 0.28}$ |
| | AT-SD | $22.59_{\pm 1.56}$ | $\underline{7.09}_{\pm 0.41}$ | $\mathbf{5.31}_{\pm 0.32}$ | $\underline{7.03}_{\pm 0.42}$ | $\mathbf{4.53}_{\pm 0.28}$ |
| | **PAT** | $\mathbf{27.26}_{\pm 0.71}$ | $\mathbf{7.40}_{\pm 0.42}$ | $\underline{5.14}_{\pm 0.31}$ | $\mathbf{7.35}_{\pm 0.48}$ | $\underline{4.08}_{\pm 0.07}$ |
| 50 | RoBal | $\underline{21.04}_{\pm 0.46}$ | $4.11_{\pm 0.47}$ | $3.11_{\pm 0.37}$ | $4.03_{\pm 0.12}$ | $2.14_{\pm 0.14}$ |
| | REAT | $\mathbf{22.12}_{\pm 0.14}$ | $3.57_{\pm 0.19}$ | $3.04_{\pm 0.20}$ | $3.53_{\pm 0.33}$ | $2.41_{\pm 0.24}$ |
| | AT-BSL | $20.31_{\pm 0.62}$ | $3.01_{\pm 0.44}$ | $2.56_{\pm 0.33}$ | $2.95_{\pm 0.13}$ | $2.09_{\pm 0.12}$ |
| | TAET | $13.16_{\pm 0.99}$ | $2.40_{\pm 0.13}$ | $1.65_{\pm 0.12}$ | $2.34_{\pm 0.09}$ | $1.34_{\pm 0.10}$ |
| | AT-SD | $17.15_{\pm 0.97}$ | $\underline{4.33}_{\pm 0.61}$ | $\underline{3.22}_{\pm 0.83}$ | $\underline{4.26}_{\pm 0.82}$ | $\mathbf{2.69}_{\pm 0.65}$ |
| | **PAT** | $20.59_{\pm 1.21}$ | $\mathbf{5.05}_{\pm 0.32}$ | $\mathbf{3.35}_{\pm 0.16}$ | $\mathbf{4.97}_{\pm 0.31}$ | $\underline{2.61}_{\pm 0.39}$ |
| 100 | RoBal | $\underline{16.63}_{\pm 0.63}$ | $2.82_{\pm 0.35}$ | $2.04_{\pm 0.45}$ | $2.75_{\pm 0.32}$ | $1.36_{\pm 0.37}$ |
| | REAT | $17.95_{\pm 0.58}$ | $2.79_{\pm 0.36}$ | $2.33_{\pm 0.03}$ | $2.73_{\pm 0.32}$ | $1.73_{\pm 0.16}$ |
| | AT-BSL | $\mathbf{16.95}_{\pm 0.52}$ | $2.40_{\pm 0.22}$ | $2.01_{\pm 0.17}$ | $2.36_{\pm 0.16}$ | $1.60_{\pm 0.18}$ |
| | TAET | $8.39_{\pm 0.73}$ | $1.29_{\pm 0.19}$ | $0.95_{\pm 0.10}$ | $1.27_{\pm 0.48}$ | $0.79_{\pm 0.16}$ |
| | AT-SD | $13.99_{\pm 0.99}$ | $\underline{3.23}_{\pm 0.23}$ | $\mathbf{2.61}_{\pm 0.11}$ | $\underline{3.21}_{\pm 0.33}$ | $\mathbf{2.31}_{\pm 0.22}$ |
| | **PAT** | $15.31_{\pm 0.28}$ | $\mathbf{3.73}_{\pm 0.31}$ | $\underline{2.45}_{\pm 0.07}$ | $\mathbf{3.68}_{\pm 0.13}$ | $\underline{1.76}_{\pm 0.31}$ |
| 200 | RoBal | $12.53_{\pm 0.59}$ | $2.00_{\pm 0.09}$ | $1.37_{\pm 0.17}$ | $1.95_{\pm 0.21}$ | $0.87_{\pm 0.10}$ |
| | REAT | $\mathbf{14.35}_{\pm 0.14}$ | $2.01_{\pm 0.22}$ | $1.59_{\pm 0.26}$ | $1.99_{\pm 0.22}$ | $1.13_{\pm 0.19}$ |
| | AT-BSL | $\underline{13.29}_{\pm 0.43}$ | $1.89_{\pm 0.20}$ | $\mathbf{1.65}_{\pm 0.01}$ | $1.85_{\pm 0.19}$ | $1.31_{\pm 0.14}$ |
| | TAET | $4.70_{\pm 0.17}$ | $0.65_{\pm 0.11}$ | $0.44_{\pm 0.06}$ | $0.65_{\pm 0.10}$ | $0.36_{\pm 0.02}$ |
| | AT-SD | $10.07_{\pm 0.50}$ | $\underline{2.09}_{\pm 0.33}$ | $\underline{1.63}_{\pm 0.31}$ | $\underline{2.09}_{\pm 0.25}$ | $\mathbf{1.36}_{\pm 0.22}$ |
| | **PAT** | $10.99_{\pm 0.37}$ | $\mathbf{2.48}_{\pm 0.32}$ | $1.46_{\pm 0.29}$ | $\mathbf{2.49}_{\pm 0.21}$ | $\underline{1.07}_{\pm 0.33}$ |

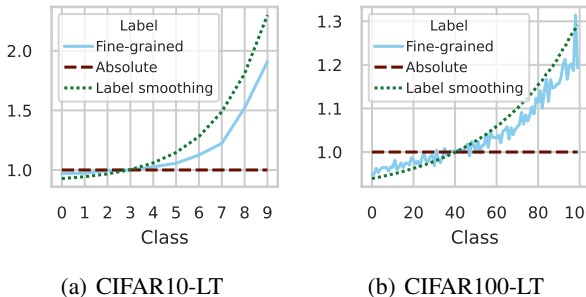

(a) CIFAR10-LT   (b) CIFAR100-LT

*Figure 8.* Class-wise ratios between class-frequency estimates obtained with different label types and the empirical class-frequency estimated from absolute labels in the dataset.

### D.2. Additional results

Additional experiment results are provided in Figs. 8 to 11 and Tables 6 to 8, the observations and analysis of which are consistent to that in the main text.

**Algorithm 1** PyTorch-style pseudocode of PAT

```
"""
Args:
    tau, r, gamma: hyper-parameters for PAT
    h: randomly initialized target model
    prior: shape=[num_classes], the class-
        frequency in dataset
Returns:
    well-trained model
"""

# surrogate initialization
q = copy.deepcopy(h)

# train posterior surrogate q for Tq epochs
for t in range(Tq):
    for (x, y) in dataloader:

        # absolute labels
        e_y = one_hot(y)

        # LSE loss
        logits = q(x) + prior.log()
        bsl_loss = - e_y * (logits - torch.
            logsumexp(logits, dim=1, keepdim=
            True))
        cls_loss = bsl_loss.mean(dim=1)
        lse_loss = tau * torch.logsumexp(
            cls_loss)

        # surrogate update
        q = update(lse_loss, q)

# train model h for T epochs
for t in range(T):

    # frequency estimate initialization
    sample_per_class = torch.zeros(num_classes)

    for (x, y) in dataloader:
        # adversarial example generation
        x_adv = adversary(x, y)

        # fine-grained labels and frequencies
        y_q, y_adv_q = softmax(q(x), dim=1),
            softmax(q(x_adv), dim=1)
        y_adv = (1 - r) * y_q + r * y_adv_q
        sample_per_class += y_adv.sum(0)

        # weight perturbation
        bsl_loss = cross_entropy(h(x_adv) +
            prior.log(), y_adv)
        wp = weight_pert(bsl_loss, h, gamma)

        # model update under weight perturbation
        h = add_into_weight(h, wp)
        bsl_loss = cross_entropy(h(x_adv) +
            prior.log(), y_adv)
        h = update(bsl_loss, h)
        h = subtract_from_weight(h, wp)

    # frequency update
    prior = sample_per_class / sample_per_class
        .sum()

# return the well-trained model
return h
```

*Table 7.* Accuracy, robustness, and training time of different variants on TinyImageNet-LT using ResNet-18. Training time is averaged per epoch. The **1st** and **2nd** results are highlighted. WP is short for weight perturbation.

| Variant | Component | | | Effectiveness | | | | Efficiency | |
|---|---|---|---|---|---|---|---|---|---|
| | Surrogate | Fine-grained Label / Freq. | WP | Acc. (all) | Acc. (worst) | Rob. (all) | Rob. (worst) | $h$ time | $q$ time |
| **PAT** | ✓ | ✓ / ✓ | ✓ | $41.16_{\pm0.16}$ | $25.23_{\pm0.53}$ | **$17.03_{\pm0.27}$** | **$6.79_{\pm0.12}$** | $3'14''$ | $37''$ |
| w/o. Surrogate | ✗ | ✗ / ✗ | ✓ | $39.95_{\pm0.38}$ | $25.09_{\pm1.12}$ | $15.21_{\pm0.25}$ | $5.67_{\pm0.14}$ | $3'14''$ | $0''$ |
| w/o. Probability | ✓ | ✗ / ✗ | ✓ | $37.84_{\pm0.26}$ | $20.46_{\pm0.43}$ | $14.67_{\pm0.12}$ | $4.63_{\pm0.35}$ | $3'14''$ | $37''$ |
| w/o. Freq. | ✓ | ✓ / ✗ | ✓ | **$41.28_{\pm0.15}$** | **$26.14_{\pm1.03}$** | $16.83_{\pm0.14}$ | $6.67_{\pm0.57}$ | $3'14''$ | $37''$ |
| w/o. WP | ✓ | ✓ / ✓ | ✗ | $39.45_{\pm0.43}$ | $25.27_{\pm0.50}$ | $13.08_{\pm0.35}$ | $4.68_{\pm0.61}$ | **$2'57''$** | $37''$ |

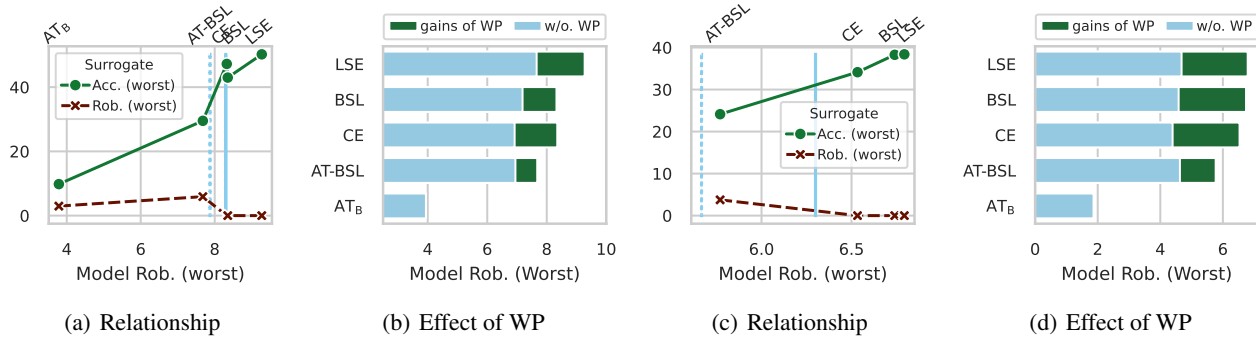

| (a) Relationship | (b) Effect of WP | (c) Relationship | (d) Effect of WP |

*Figure 9.* Relationship between posterior surrogate accuracy and resulting model robustness of worst-class, and the effect of Weight Perturbation (WP) on this relationship. (a) and (b): results on CIFAR100-LT. (c) and (d): results on TinyImageNet-LT. Blue lines in (a) and (c) correspond to using absolute labels without (dashed) and with (solid) label smoothing. With WP stabilizing the sensitivity to approximation error of surrogate, surrogate accuracy reliably translate into resulting robustness of the target model.

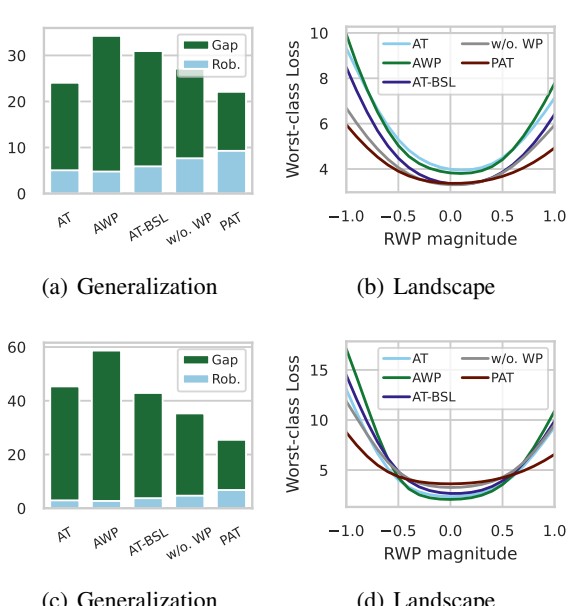

| (a) Generalization | (b) Landscape |
| (c) Generalization | (d) Landscape |

*Figure 10.* Relationship between weight loss landscape and robust generalization gap on worst-class. The landscape is visualized by Random Weight Perturbation (RWP) in (Wu et al., 2020). 'w/o. WP' denotes the variant of PAT that removes the weight perturbations. With fine-grained supervisions, training with weight perturbations yields a flatter loss landscape and plays a critical role in reduce robust generalization gap.

*Table 8.* Accuracy of posterior surrogates under different $\tau \in \{1, 10, 100\}$ using ResNet-18. The **1st** results are highlighted. We observe that the optimal $\tau$ scales proportionally with the number of classes $C$, which is 10, 100, and 200 for CIFAR10-LT, CIFAR100-LT, and TinyImageNet-LT, respectively. We therefore suggest $\tau = 10^{\lceil \lg C \rceil - 1}$ on the these benchmarks.

| Dataset | Acc. | $\tau = 1$ | $\tau = 10$ | $\tau = 100$ |
|---|---|---|---|---|
| CIFAR10-LT | all | **$83.97_{\pm0.17}$** | $77.33_{\pm1.46}$ | $10.00_{\pm0.00}$ |
| | worst | **$77.13_{\pm0.58}$** | $67.77_{\pm1.48}$ | $0.00_{\pm0.00}$ |
| CIFAR100-LT | all | $44.29_{\pm0.59}$ | **$64.16_{\pm0.23}$** | $61.08_{\pm0.65}$ |
| | worst | $10.62_{\pm1.51}$ | **$50.21_{\pm0.64}$** | $46.95_{\pm0.15}$ |
| TinyImageNet-LT | all | $0.50_{\pm0.00}$ | $46.81_{\pm0.34}$ | **$53.52_{\pm0.28}$** |
| | worst | $0.00_{\pm0.00}$ | $30.71_{\pm0.88}$ | **$38.38_{\pm0.66}$** |

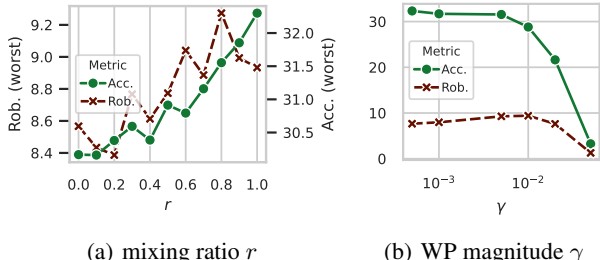

| (a) mixing ratio $r$ | (b) WP magnitude $\gamma$ |

*Figure 11.* Sensitivity of PAT to hyper-parameters $r \in [0.0, 1.0]$ and $\gamma \in [5 \times 10^{-4}, 5 \times 10^{-2}]$ on CIFAR100-LT using ResNet-18. Results on worst-class are reported. Moderate $r$ and $\gamma$ improve robustness, while excessively large ones hurt robustness.

