# OpenReview forum: "Posterior Mismatch Matters: Adversarial Training for Long-Tailed Robustness"
_ICML.cc/2026/Conference — ICML 2026 regular_

### Official Review · Reviewer_Qptr · 2026-03-02

**Soundness:** 3
**Presentation:** 3
**Significance:** 2
**Originality:** 2
**Overall Recommendation:** 4
**Confidence:** 3

**Summary:**

This work demonstrates that coarse-grained absolute labels tend to collapse class posteriors into point estimates in the adversarial long-tailed setting, resulting in an enlarged robust generalization gap. To address this issue, the authors propose using fine-grained probabilistic labels and incorporating weight perturbations to encourage a flatter loss landscape. Extensive experiments show that the proposed method achieves state-of-the-art performance.

**Compliance With Llm Reviewing Policy:**

Affirmed.

**Final Justification:**

I will maintain my current score.

**Key Questions For Authors:**

- Theorems 4.5 and 4.9 appear to be extendable to the non-adversarial long-tailed setting. Are there prior non-adversarial works that derive similar generalization gap results? A discussion of related theoretical results would help contextualize your contribution.
- The pair-wise loss, e.g., AUC, can deal with the class-imbalanced dataset, and there are several relevant works, e.g., [1]. Could you provide either empirical or conceptual comparisons between your method and such pairwise-loss-based approaches?

[1] Hou, W., Xu, Q., Yang, Z., Bao, S., He, Y., & Huang, Q. (2022, June). Adauc: End-to-end adversarial auc optimization against long-tail problems. In International Conference on Machine Learning (pp. 8903-8925). PMLR.

**Limitations:**

Yes

**Strengths And Weaknesses:**

Strengths
- This paper is well-structured and easy to follow. The motivation and contributions are clear.
- The theoretical analysis (e.g., the theorem characterizing the robust generalization gap for absolute labels versus approximated probabilistic labels) provides useful insight into why probabilistic labels may be advantageous.

Weaknesses
- Some equations and derivations are difficult to follow.
    - In Eq. (8), it is unclear why the variable $u$ disappears on the right-hand side and why the maximum is taken over all classes.
    - In Theorem 4.5(Line 193), the $u$ does not appear in $\mathcal{R}(h,p)$, and it is confusing to take the expectation over $\mathcal{U}$.
- In Line 206, the authors state that “existing methods based on absolute labels typically do not explicitly control sharpness.” However, Adversarial Weight Perturbation (AWP) appears to explicitly regularize sharpness. The distinction between your method and such approaches should be clarified.
- It seems that Theorems 4.5 and 4.9 may also hold in adversarial but non-long-tailed settings. The paper would benefit from a clearer explanation of why probabilistic labels are particularly necessary or beneficial in the long-tailed regime compared to the standard adversarial setting.

---

> ### Author Rebuttal · Authors · 2026-03-31
>
> We are grateful to Reviewer Qptr for the careful reading and insightful suggestions.
>
> ### Weakness 1: Equation typos
>
> We thank the reviewer for pointing this out. The correct form should be $\mathbb{E}_{u \in \mathcal{U}} [ \mathcal{R}(h+u, p) ]$ in Eq. (8). The variable $u$ added to $h$ on the left-hand-side in Theorem 4.5 (and 4.9) was missing. We will carefully proofread the paper and correct all typos in the final version.
>
> ### Weakness 2: Distinction to AWP
>
> We clarify that “existing methods” here refer to long-tailed adversarial training approaches, which mainly focus on imbalance-aware objectives and typically do not explicitly control sharpness. While approaches such as AWP regularize sharpness, they are designed for class-balanced settings. In contrast, we study the intersection of long-tail and adversarial robustness, where posterior mismatch couples posterior estimation error with weight loss landscape sharpness. Our method explicitly integrates posterior approximation with weight perturbation to address this issue. We will clarify this distinction in the revision.
>
> ### Weakness 3: Necessity in long-tailed regime
>
> We agree that Th. 4.5 & Th. 4.9 can extend to class-balanced adversarial settings. Indeed, posterior mismatch is a general phenomenon that affects both adversarial and long-tailed regimes. To better isolate these effects, we explicitly decompose our analysis in Sec. 4.1 into: (i) long-tailed settings (Th. 4.2 & Th. 4.7), and (ii) adversarial settings (Th. 4.5 & Th. 4.9), which lead to different subsets of our theoretical results.
>
> **(1) In class-balanced adversarial settings**, posterior mismatch mainly increases loss variance and landscape sharpness, affecting robust generalization.
>
> **(2) In long-tailed adversarial settings**, posterior mismatch additionally induces biased class-frequency estimation, which directly impacts rebalancing mechanisms (e.g., BSL) that rely on class priors, thereby harming long-tailed robustness.
>
> Therefore, while probabilistic labels for fixing posterior mismatch are beneficial more broadly, they become particularly important in long-tailed adversarial settings due to this additional source of bias. We will clarify this distinction in the revision.
>
> ### Question 1: Related theoretical results
>
> Our contribution of Th. 4.5 & Th. 4.9 differs in focusing on the long-tailed adversarial setting, where the effects of (i) sample variance and (ii) landscape sharpness are intrinsically coupled.
>
> Prior theoretical works study these two factors separately. For instance, an existing work (Menon et al., 2021) derives generalization gap with sample variance of loss to show that why soft labels from a teacher is beneficial for general knowledge distillation. Another work (Wu et al., 2020) derives robust generalization gap that considering landscape sharpness for standard adversarial settings, showing how flatenss controlling improves the gap.
>
> Our results suggest that the accurate posterior approximation can reduce the former, while flatness controlling reduces the latter. Importantly, flatness controlling also stabilizes the effect of approximation error, ensuring that improvements in posterior estimation translate into consistent long-tailed robustness gains. We will clarify this distinction and related discussion in the revision.
>
>
> ### Question 2: Pair-wise loss baseline
>
> We thank the reviewer for highlighting this relevant line of work. Pairwise AUC optimization methods (e.g., AdAUC) are effective for class imbalance via ranking objectives.
>
> **(1) Conceptual differences.** (i) AUC-based methods optimize pairwise ordering and still rely on class-frequency priors derived from absolute labels. They do not explicitly model class posteriors, and thus remain affected by posterior mismatch. In contrast, our method directly solving posterior mismatch by exploiting posterior approximation, which we show is critical for long-tailed robust generalization. (ii) AUC is most natural for binary tasks and requires decomposition for multi-class settings. Our method is inherently multi-class and integrates directly with standard adversarial training.
>
> **(2) Empirical comparison.**  We reproduce AdAUC with its original hyper-parameter settings and extend it to multi-class via pairwise averaging over all class pairs. Acc./Rob. results on worst-class are shown as follows:
>
> | Method (Metric)   | CIFAR10-LT    | CIFAR100-LT  |
> | ----------------- | ------------- | ------------ |
> | AdAUC (Acc./Rob.) | 44.77 / 10.23 | 19.13 / 1.15 |
> | PAT (Acc./Rob.)   | 64.03 / 26.23 | 31.55 / 9.27 |
>
> While AdAUC improves imbalance handling, it is less effective than PAT, suggesting that pairwise objectives alone do not fully address posterior mismatch in adversarial multi-class settings. Exploring combinations of pairwise losses with our posterior modeling framework is an interesting direction for future work. We will include this discussion in the revision.

---

> > ### Author Rebuttal · Reviewer_Qptr · 2026-04-01
> >
> > My concerns have been adequately addressed. However, I will maintain my current score, since the proposed Posterior-driven Adversarial Training seems somewhat incremental in nature, as it mainly builds on BSL and AWP.

---

> > > ### Author Response · Authors · 2026-04-01
> > >
> > > Thank you for the positive feedback and for confirming that your concerns have been addressed. We will diligently continue to refine the paper and ensure that the additional results and discussion are incorporated.
> > >
> > > We respectfully note that our contribution goes beyond a direct combination of existing components such as BSL and AWP.
> > >
> > > The key novelty lies in identifying **posterior mismatch** as a root cause of failure in long-tailed adversarial training, and in providing a **unified theoretical analysis** in Section 4 that connects posterior approximation and weight loss landscape sharpness. Based on this analysis, we show that posterior approximation and flatness control are **intrinsically coupled** in Theorem 4.9, rather than independent design choices. This insight motivates PAT as a principled framework where the surrogate (for posterior estimation) and weight perturbation (for flatness) work together to improve worst-class robustness.
> > >
> > > Regarding BSL, although we adopt it as the training objective, prior works largely treat it as a heuristic. In contrast, we provide **the first theoretical analysis of the BSL** in Lemma 3.1, including its theoretical properties and optimal hyperparameter choice, offering a principled understanding of when and why it is effective.
> > >
> > > We will further emphasize these conceptual and theoretical contributions in the final version.

---

### Official Review · Reviewer_VUrt · 2026-03-05

**Soundness:** 3
**Presentation:** 3
**Significance:** 3
**Originality:** 3
**Overall Recommendation:** 4
**Confidence:** 3

**Summary:**

This paper studies the degradation of adversarial training in long-tailed recognition settings and attributes it to a posterior mismatch between one-hot labels and adversarially perturbed samples. To address this, this paper proposes Posterior-driven Adversarial Training (PAT), which utilizes soft probabilistic labels to replace absolute labels during adversarial training. Extensive experiments across three datasets and multiple model architectures demonstrate that the proposed method improves both model accuracy and adversarial robustness.

**Compliance With Llm Reviewing Policy:**

Affirmed.

**Final Justification:**

The paper offers a novel perspective on long-tailed adversarial training by attributing its performance degradation to label posterior mismatch, and theoretically demonstrates that absolute labels bias class-frequency estimation, thereby enlarging the robust generalization gap. The authors’ response has adequately addressed my concerns, and I am therefore inclined to accept this paper.

**Key Questions For Authors:**

see weaknesses

**Limitations:**

Yes

**Strengths And Weaknesses:**

### Strengths

- The paper provides a valuable perspective by analyzing the performance degradation of long-tailed adversarial training through the lens of label posterior mismatch.
- The proposed method is supported by theoretical analysis detailing how absolute labels induce bias in class-frequency estimation and enlarge the robust generalization gap.
- The paper compares the proposed method against various adversarial training baselines across multiple datasets and model architectures.

### Weaknesses

- The surrogate model $q$ is trained only on clean data, but it is required to produce outputs on adversarial samples $\tilde{x}$ when constructing soft labels. It is unclear whether $q(\tilde{x})$ can reliably reflect the posterior shift under adversarial perturbations, especially under stronger attacks.

- The principle behind the construction of the soft labels (Eq. 13) is not sufficiently explained. In particular, it is unclear why the soft label from the original image is necessary.

- The experimental results suggest that the mixing ratio $r$ has a significant impact on both accuracy and robustness, and its effect is not consistent across datasets (e.g., Figures 5a and 10a). However, this paper does not provide a clear guideline for selecting an appropriate value of $r$.

- The experiments mainly compare different adversarial training methods, but a standard natural training baseline is missing. This makes it difficult to quantify the impact of adversarial training on clean accuracy.

- There are several minor typos and wording issues in the paper (e.g., "simplely" -> "simply").

---

> ### Author Rebuttal · Authors · 2026-03-31
>
> We sincerely appreciate Reviewer VUrt’s careful review and invaluable feedback.
>
> ### Weakness 1: Posterior shift reflecting
>
> We understand the reviewer’s concern. Although the surrogate $q$ is trained only on clean data and is not adversarially robust, the mixed labels can capture meaningful posterior shifts under perturbations.
>
> **(1) Capturing posterior shift.** The lack of adversarial robustness in surrogate $q$ is in fact beneficial. An adversarially trained $q$  would tend to enforce invariance and map adversarial data to the same absolute label as clean data, thereby suppressing posterior shifts. In contrast, a non-robust $q$ remains sensitive to adversarial perturbations and can reflect fine-grained changes in class probabilities induced by them.
>
> **(2) Reliability of the estimated shift.** Adversarial samples are generated against the target model $h$, not $q$, so their transferability to $q$ is limited. Empirically, we report Acc./Rob./A-Acc. (accuracy on adversarial samples against $h$ at the last epoch) of $q$:
>
> | Metric           | CIFAR10-LT           | CIFAR100-LT          | TinyImageNet-LT      |
> | ---------------- | -------------------- | -------------------- | -------------------- |
> | Acc./Rob./A-Acc. | 87.23 / 0.01 / 62.06 | 75.69 / 0.01 / 53.96 | 69.36 / 0.01 / 53.98 |
>
> The relatively high A-Acc. (vs. near-zero Rob.) indicates that $q$ still preserves meaningful predictive structure on adversarial data. In addition, incorporating clean soft labels in Eq. (13) regularizes the mixed label and prevents overestimating the shift.
>
> Overall, our method combines sensitive (adversarial) and stable (clean) predictions to reliably capture posterior changes. We will clarify this in our work.
>
> ### Weakness 2: Design of Eq. (13)
>
> **(1) Principle behind Eq. (13).** The design of Eq. (13) is motivated by two considerations: (i) *Consistency for robustness:* adversarial robustness requires the model to produce consistent predictions for clean and adversarial inputs; (ii) *Posterior shift under perturbations:* although the absolute label remains unchanged, adversarial perturbations can alter the underlying class posterior. Accordingly, we mix the surrogate predictions on clean and adversarial inputs. The clean component serves as a stable reference that enforces prediction consistency, while the adversarial component captures local posterior shift caused by adversarial perturbation.
>
> **(2) Necessity of clean soft label for robustness.** Using only adversarial soft label tends to reduce training to a form of distillation and breaks consistency between clean and adversarial supervision, leading to degraded robustness. Introducing the clean soft label enables the model to better address robustness in long-tailed scenarios under posterior mismatch. We will clarify the necessity of this combination in the paper.
>
> ### Weakness 3: Impact of the mixing ratio $r$
>
> We agree that $r$ affects both accuracy and robustness. We observe a similar trend across datasets:
>
> **(1) Robustness** typically follows a unimodal pattern: it improves as $r$ increases from small values, but degrades when $r$ becomes too large. Moderate $r$ helps capture local posterior structure around adversarial data that helps mitigate posterior mismatch. When $r$ is large (e.g., $r=1$), supervision is dominated by adversarial soft label, weakening consistency between clean and adversarial data and harming robustness.
>
> **(2) Accuracy** generally increases with $r$, as the objective increasingly aligns the prediction behaviors of the target model with that of the surrogate. This resembles distillation and transfers the surrogate’s standard generalization, as observed in Fig. 5(a) and Fig. 10(a).
>
> **(3) As a guideline**, we set $r=0.6$ for CIFAR10-LT and $r=0.8$ for CIFAR100-LT and TinyImageNet-LT, selected based on validation robustness performance and observed to generalize well across architectures. In practice, $r=0.6$ serves as a robust default when hyperparameter tuning is limited.
>
> ### Weakness 4: Standard training baseline
>
> We focus on long-tailed robustness, and thus mainly compare against adversarial training baselines. We agree that including a standard training baseline helps contextualize the trade-off between accuracy and robustness. We report worst-class Acc./Rob. as follows:
>
> | Method (Metric)      | CIFAR10-LT    | CIFAR100-LT  | TinyImageNet-LT |
> | -------------------- | ------------- | ------------ | --------------- |
> | standard (Acc./Rob.) | 65.69 / 0.00  | 42.97 / 0.00 | 34.13 / 0.00    |
> | PAT (Acc./Rob.)      | 64.03 / 26.23 | 31.55 / 9.27 | 25.23 / 6.79    |
>
> As expected, standard training achieves higher accuracy but near-zero robustness, while PAT significantly improves robustness with a moderate trade-off in accuracy. We will include this baseline for completeness.
>
> ### Weakness 5: Typos
>
> We thank the reviewer for pointing this out. We will carefully proofread the paper and correct all typos in the final version.

---

> > ### Author Rebuttal · Reviewer_VUrt · 2026-04-01
> >
> > My concerns have been well addressed, and I will increase the score.

---

> > > ### Author Response · Authors · 2026-04-01
> > >
> > > We sincerely appreciate your encouraging feedback and strong support. Your valuable insights are very helpful, and we are carefully integrating all clarifications and additional results into the updated manuscript.

---

### Official Review · Reviewer_4Rht · 2026-03-08

**Soundness:** 3
**Presentation:** 3
**Significance:** 3
**Originality:** 3
**Overall Recommendation:** 5
**Confidence:** 2

**Summary:**

This paper studies adversarial training under long-tailed distributions, and show that when they coexist performance decreases especially on worst-performing classes. The paper argues that a key reason is a posterior mismatch, using absolute labels collapses the class posterior into a point estimate, which can bias class-frequency estimation and enlarge the robust generalization gap. They propose Posterior-driven Adversarial Training (PAT): (i) learn a posterior surrogate q on clean data, (ii) use q to build fine-grained soft labels and also update class frequencies using these soft labels, and (iii) add weight perturbations to encourage a flatter weight loss landscape and reduce sensitivity to posterior approximation errors. Experiments on CIFAR10-LT, CIFAR100-LT and TinyImageNet-LT show consistent gains, with large improvements on worst-class robust accuracy.

**Compliance With Llm Reviewing Policy:**

Affirmed.

**Final Justification:**

My final recommendation is accept. The paper tackles an important problem in long-tailed adversarial training and provides a technically solid method with clear motivation, helpful ablations, and strong improvements, especially on worst-class robust accuracy.

My main concerns were about the practical stability of the surrogate-based class-frequency updates and the reproducibility of the full pipeline given the added surrogate and hyperparameters. The rebuttal addressed these satisfactorily by clarifying that the surrogate is pretrained and frozen, providing recommended default settings, and discussing training stability more explicitly.

Overall, the rebuttal improved the clarity of the method and resolved the main practical issues I had raised, so I increased my score to accept.

**Key Questions For Authors:**

1-) How do you ensure this update is stable in practice ( do you use any smoothing, clipping, or warm-up), and did you observe oscillations or drift of $\hat{f}^q$ across training?

2-) Is the posterior surrogate q trained once pretrained on clean data and then frozen, or updated jointly during adversarial training? If frozen, do you ever refresh it after h changes?

3-) Do you have a way of telling whether q is "good enough" especially on the worst-class? For example, do you monitor worst-class surrogate accuracy and use it to decide hyperparameters like r or γ?

**Limitations:**

yes

**Strengths And Weaknesses:**

Strengths

Soundness: The paper gives a clear theoretical motivation. Absolute labels increase variance in class-frequency estimation and can worsen robust generalization bounds.

Presentation: The main idea is explained clearly, and the method section is structured. The ablations are also helpful.

Significance: Worst-performing class robustness in long-tailed settings is practically important, since attacks can target the vulnerable tail classes, and improving robustness beyond average metrics is meaningful.

Originality: The posterior mismatch framing plus using a learned posterior surrogate as supervision for long-tailed adversarial training is a reasonably novel angle.

Weaknesses

1-) Since class-frequency prior $\hat{f}^q$ is computed from the surrogate-produced soft labels, errors in q can directly affect the rebalancing. You already discuss approximation error in theory, however I am more concerned about the general stability of this $\hat{f}^q$ updates.

2-) PAT introduces an extra model q, new hyperparameters (r, ρ, $\gamma$), and frequency updates, plus WP. Even if each part is motivated, it increases the number of knobs and makes the pipeline harder to reproduce and tune. It would help to clearly summarize what needs tuning vs what is stable across datasets, and what the authors recommend as default settings.

---

> ### Author Rebuttal · Authors · 2026-03-31
>
> We thank Reviewer 4Rht for raising important concerns and questions, which have helped improve the clarity of our work.
>
> ### Weakness 1: Stability of class-frequency update
>
> We agree that approximation error in surrogate $q$ could affect the rebalancing mechanism via fine-grained class-frequency update, as discussed theoretically. We argue that this design guarantees rather than destroys the general stability.
>
> **(1) Stable objective.** The class-frequency update avoids a distributional inconsistency in training objective, which could otherwise lead to unstable optimization. Optimizing the BSL objective with respect to a data-label distribution encourages the model $h$ to capture the mutual information between data and label as shown in Lemma 3.1. Therefore, training with the fine-grained class-frequency distributionally consistent with the fine-grained labels ensures a stable optimization objective.
>
> **(2) Stable class-frequency.** The fine-grained labels are relatively stable, as the surrogate $q$ produces them for the data on which it was trained, thereby stablizing the fine-grained class-frequency estimate for update. This empirically leads to well-calibrated predictions, especially for clean samples. While adversarial examples adding perturbations to clean samples, these perturbations are worst-case for the target model $h$ rather than $q$, whose transferability to $q$ is limited. Empirically, we report Acc./Rob./A-Acc. (accuracy on adversarial samples against $h$):
>
> | Metric           | CIFAR10-LT           | CIFAR100-LT          | TinyImageNet-LT      |
> | ---------------- | -------------------- | -------------------- | -------------------- |
> | Acc./Rob./A-Acc. | 87.23 / 0.01 / 62.06 | 75.69 / 0.01 / 53.96 | 69.36 / 0.01 / 53.98 |
>
> The relatively high A-Acc. (vs. near-zero Rob.) indicates that $q$ still preserves meaningful predictive structure on adversarial inputs. Therefore, these labels remain relatively reliable, which in turn stabilizes the class-frequency estimates derived from them.
>
> **(3) Empirical perspective.** Ablation results in Table 5 show that incorporating the fine-grained frequency update consistently improves worst-class robustness, without enlarging the standard deviation.
>
> Overall, the fine-grained class-frequency update leads to stable training behavior in practice. We will clarify this point in the paper.
>
> ### Weakness 2: Recommended hyper-parameter settings
>
> PAT introduces a surrogate and weight perturbation. Most of its hyper-parameters are stable across different settings and only require minimal tuning in practice. We summarize our recommended settings as follows:
>
> - $\tau$ (LSE temperature). We set $\tau = 10^{\lceil \log C \rceil - 1}$ to ensure it scales with the number of classes $C$. This choice works consistently across all datasets in our experiments and does not require further tuning.
> - $\gamma$ (weight perturbation strength). We use $\gamma = 5 \times 10^{-3}$, following prior work AWP and also validated in our experiments. This choice is stable across datasets and architectures.
> - $r$ (mixing ratio). Robustness typically peaks at an intermediate value within $r \in (0,1)$. We set $r=0.6$ for CIFAR10-LT and $r=0.8$ for CIFAR100-LT and TinyImageNet-LT, selected based on validation robustness and observed to generalize well across architectures. In practice, $r=0.6$ serves as a robust default when hyperparameter tuning is limited.
> - Other hyper-parameters (e.g., learning rate) follow standard practices without additional sensitivity.
>
> Overall, PAT does not require extensive tuning: most parameters are either fixed by design (e.g., $\tau$) or robust across different settings (e.g., $\gamma$), and only $r$ may benefit from light tuning. We will include these default settings and sensitivity discussions in the paper to improve reproducibility.
>
> ### Question 1: Stability of training
>
> We do not employ additional techniques. Importantly, we do not observe abnormal oscillations or drift during training. This stability is mainly attributed to the design of PAT, as supported by our analysis in the response to Weakness 1. We will further clarify this observation by including learning curves in the paper.
>
> ### Question 2: State of surrogate $q$
>
> The posterior surrogate $q$ is pretrained on clean data and then kept fixed during adversarial training, without further updates or refreshing. This design decouples posterior estimation from adversarial optimization, improving stability while avoiding the complexity of joint training.
>
> ### Question 3: Quality guarantee for surrogate $q$
>
> We do not explicitly monitor surrogate quality for adaptive tuning. Instead, we rely on the surrogate training objective, which emphasizes worst-class performance via the LogSumExp formulation in Eq. (11). Empirically, this objective produces sufficiently accurate and well-calibrated predictions for downstream use, as reflected in Fig. 3(a), Fig. 8(a) and Fig. 8(c).

---

> > ### Author Rebuttal · Reviewer_4Rht · 2026-04-01
> >
> > Thank you for the clear rebuttal. The authors addressed several of my practical questions helpfully, in particular by clarifying that the posterior surrogate is pretrained and then frozen, and by providing recommended default settings for the main hyperparameters. I also appreciate the added discussion of stability and the claim that no abnormal oscillation or drift was observed in practice.

---

> > > ### Author Response · Authors · 2026-04-01
> > >
> > > We are very grateful for your encouraging feedback and strong support. We will diligently continue to refine the paper and ensure that the discussions regarding training stability and hyper-parameter guidelines are incorporated.

---

### Official Review · Reviewer_VPGt · 2026-03-09

**Soundness:** 2
**Presentation:** 2
**Significance:** 2
**Originality:** 2
**Overall Recommendation:** 4
**Confidence:** 5

**Summary:**

The paper proposes the posterior-driven adversarial training (PAT) to account for minor classes in classification tasks with a long-tail distribution. It trains a second neural network using group DRO denoted as q, and uses the q distribution as a soft label to do adversarial training. The q distribution also serves as a basis to accumulate frequency for each class along the training. On several datasets the proposed PAT achieves the best performance compared with recent baselines.

**Compliance With Llm Reviewing Policy:**

Affirmed.

**Final Justification:**

The response has addressed most of my concerns so I have increased my score from 3 to 4.

**Key Questions For Authors:**

Why does increasing r in Equation 13 improve the accuracy as shown in Fig. 5a? It seems that when r=1, q becomes the output of a perturbed input, which should be quite noisy.

**Limitations:**

No. See the weaknesses on presentation and generalizability of the proposed method.

**Strengths And Weaknesses:**

Strengths:

1.	Improving the adversarial robustness in long-tail class distributions is challenge task and the performance of PAT looks good.
2.	The experiment looks comprehensive.




Weaknesses:

1.	The concept of q distribution is a bit confusing here. Following Equation 11, q is defined as the arg min of the LSE max over q. It is unclear why and how the min-max problem is introduced. Intuitively, the min-max problem introduced in Equation 11 is essentially a group distributionally robust optimization (group DRO) problem, which produces a model that respects the class distribution. Then, this distribution is used later as a soft label for the class prior in Equation 13-15. Feel like the logic here is not entirely coherent to me.
2.	Related to Weakness 1, some key design choices are not fully justified. For instance, the mixing of the two q’s in Equation 13 and the frequency update in Equation 14.
3.	It is not immediately clear how the AWP is related to the long-tail adversarial robustness problem, seems that the AWP is orthogonal to the first contribution.
4.	Related to Weakness 3, it looks like AWP is indispensable for robustness of worse classes in Table 5, which may indicate that the proposed group DRO distillation is not generalizable to other AT settings.

---

> ### Author Rebuttal · Authors · 2026-03-31
>
> We are grateful to Reviewer VPGt for the helpful and insightful comments.
>
> ### Weakness 1: Concept of the surrogate $q$
>
> We appreciate the reviewer’s comment and clarify as follows:
>
> **(1) Min-max formulation.** Eq. (11) is designed to obtain a posterior approximation that remains accurate even on the worst-class as required by Guideline 1 in Sec. 4, which is not intended as a group DRO objective. The LSE loss emphasizes high-loss classes, and $g$ minimizes it to obtain uniformly good performance across classes, leading to better posterior approximation.
>
> **(2) Role of $q$.** The learned surrogate $q$ provides fine-grained probabilistic labels that better approximate Bayesian labels. This mitigates the negative effects of posterior mismatch caused by absolute labels as shown in Effects 1 & 2 in Sec. 4, thereby improving robust generalization in long-tailed settings.
>
> ### Weakness 2: Motivation of the design
>
> We clarify the key design choices as follows:
>
> **(1) Frequency update.** In long-tailed settings, class-frequency estimation is required as a prior for rebalancing techniques such as BSL. However, under posterior mismatch, class-frequency estimates from absolute labels are biased as shown in Effect 1 in Sec. 4. The surrogate $q$'s accurate approximation of class posteriors enables more accurate class-frequency estimation as shown in Guideline 1 in Sec. 4, thereby motivating Eq. (14).
>
> **(2) Mixing clean and adversarial posteriors.** The construction in Eq. (13) is motivated by two considerations: (i) *Consistency for robustness:* adversarial robustness requires the model to produce consistent predictions for clean and adversarial inputs; (ii) *Posterior shift under perturbations:* although the absolute label remains unchanged, adversarial perturbations can alter the underlying class posterior. Accordingly, we mix the surrogate predictions on clean and adversarial inputs. The clean component serves as a stable reference that enforces prediction consistency, while the adversarial component captures local posterior shift caused by adversarial perturbation. This combination enables the model to better account for robustness in long-tailed scenarios under posterior mismatch.
>
> ### Weakness 3 & Weakness 4: Role of weight perturbations
>
> We understand the reviewer’s concern and clarify as follows.
>
> Weight perturbation (WP) is not orthogonal to addressing posterior mismatch (the first contribution). Instead, it plays a complementary and crucial role in addressing the robust generalization gap induced by posterior mismatch.
>
> **(1) Theoretical perspective.** Our analysis (Th. 4.9) shows that the robust generalization gap depends on two coupled factors: (i) posterior approximation error and (ii) weight loss landscape sharpness. The posterior surrogate reduces the former, while WP reduces the latter by encouraging flatter minima. Importantly, WP also stabilizes the effect of approximation error, ensuring that improvements in posterior estimation translate into consistent long-tailed robustness gains.
>
> **(2) Empirical evidence.** As shown in Fig. 3(b), (i) with WP, improved surrogate quality reliably leads to better robustness; (ii) without WP, this relationship becomes unstable. Moreover, using WP alone without the posterior surrogate leads to suboptimal performance in Table 5. These results indicate that the two components are complementary rather than independent.
>
> **(3) Generality of the framework.** The necessity of WP does not indicate limited generality of our method. Instead, it reflects that long-tailed robust generalization inherently requires both accurate posterior estimation and flatness control. WP serves as a general mechanism for the flatness control, and our framework which combines posterior surrogate and WP can be integrated with other adversarial training losses.
>
> We will clarify this relationship in the paper.
>
> ### Question: Effect of mixing coefficient $r$ on accuracy
>
> As $r$ increases, the mixed label in Eq. (13) is increasingly dominated by the surrogate predictions on adversarial samples $q(\tilde{x})$. In the extreme case where $r=1$, the objective reduces to aligning the target model $h$ with surrogate $q$ on adversarial inputs, resembling knowledge distillation. Since $q$ is trained on clean data and has good standard generalization, this alignment transfers its predictive behavior to $h$, thereby improving natural accuracy, as observed in Fig. 5(a) and Fig. 10(a).
>
> However, $q(\tilde{x})$ still provides meaningful posterior shift information rather than pure noise when $r > 0$. Setting $r=1$ (i.e., relying solely on $q(\tilde{x})$) weakens consistency between clean and adversarial supervision, which is crucial for robustness. A moderate $0< r < 1$ balances capturing posterior shifts and preserving consistency, explaining why accuracy increases with $r$ while robustness peaks at an intermediate value in Fig. 5(a) and Fig. 10(a). We will clarify this trade-off in the paper.

---

> > ### Author Rebuttal · Reviewer_VPGt · 2026-04-01
> >
> > Thanks for the response. Some of my concerns are addressed but I am still not sure about the difference between the proposed method and group DRO. Given the description "The LSE loss emphasizes high-loss classes, and $g$ minimizes it to obtain uniformly good performance across classes, leading to better posterior approximation", it looks like this procedure is similar to group DRO. Besides, what is $g$ here? Is it supposed to be $q$?

---

> > > ### Author Response · Authors · 2026-04-01
> > >
> > > Thank you for the positive feedback that some concerns are addressed and the insightful follow-up question. We apologize for the confusion in notation. $g$ should indeed be $q$.
> > >
> > > We agree that if each class is treated as a group, Eq. (11) bears a formal similarity to group DRO since both emphasize high-loss groups and encourage uniform performance across all groups.
> > >
> > > Our reason for stating that ‘Eq. (11) is not intended as a group DRO objective’ in the initial response is the following. While Eq. (11) resembles a classification objective, the surrogate $q$ does not serve a classification purpose. When training surrogate $q$, only absolute labels are available and they serve solely as empirical supervision for approximating the posterior. Precisely because only absolute labels are accessible, each sample can be definitively assigned to a class. This, we believe, is the key reason each class can be viewed as a group in Eq. (11), resulting in the similarity. Hence, we contend that Eq. (11) is not intended as a group DRO objective.
> > >
> > > We will clarify this point in the revised version.

---

### Decision · Program_Chairs · 2026-04-30

**Decision:**

Accept (regular)

**Comment:**

The submission considers the problem of adversarial robustness in the case of long-tail class distributions. The submission theoretically analyses the problem by investigating how absolute labels (the argmax of the labeller's belief) can lead to error when estimating the class frequency. This is followed by more thorough investigation into the robust generalisation gap. A novel method is proposed on the back of this analysis, and extensive experimental evaluation shows that it performs well compared to existing approaches. Additional experiments corroborate the mechanism for the success of the method.

Several issues were identified in the original reviews, primarily related to clarity and completeness of explanations, with some additional questions relating to the stability of the proposed approach. However, many of the points raised were in favour of the submission: it investigates an important problem, provides an original and effective solution, and appears to be technically sound. The presentation issues were largely resolved during the course of the rebuttal, and it does not appear that the stability issues are a substantial concern in practice. For this reason, I recommend accepting the paper.